# Revealing concealed cardioprotection by platelet Mfsd2b-released S1P in human and murine myocardial infarction

Amin Polzin[1,2,3,6], Lisa Dannenberg[1,6], Marcel Benkhoff [1,6], Maike Barcik[1], Carolin Helten[1], Philipp Mourikis[1], Samantha Ahlbrecht[1], Laura Wildeis[1], Justus Ziese[1], Dorothee Zikeli[1], Daniel Metzen[1], Hao Hu[1], Leonard Baensch[1], Nathalie H. Schröder[4], Petra Keul[4], Sarah Weske[4], Philipp Wollnitzke[4], Dragos Duse[1,4], Süreyya Saffak[1], Mareike Cramer[1], Florian Bönner[1,2], Tina Müller[5], Markus H. Gräler[5], Tobias Zeus[1,2], Malte Kelm[1,2] & Bodo Levkau [4] ✉

Antiplatelet medication is standard of care in acute myocardial infarction (AMI). However, it may have obscured beneficial properties of the activated platelet secretome. We identify platelets as major source of a sphingosine-1-phosphate (S1P) burst during AMI, and find its magnitude to favorably associate with cardiovascular mortality and infarct size in STEMI patients over 12 months. Experimentally, administration of supernatant from activated platelets reduces infarct size in murine AMI, which is blunted in platelets deficient for S1P export (*Mfsd2b*) or production (*Sphk1*) and in mice deficient for cardiomyocyte S1P receptor 1 (S1P$_1$). Our study reveals an exploitable therapeutic window in antiplatelet therapy in AMI as the GPIIb/IIIa antagonist tirofiban preserves S1P release and cardioprotection, whereas the P2Y12 antagonist cangrelor does not. Here, we report that platelet-mediated intrinsic cardioprotection is an exciting therapeutic paradigm reaching beyond AMI, the benefits of which may need to be considered in all antiplatelet therapies.

Platelet activation on ruptured or eroded atherosclerotic lesions leads to thrombus formation and ischemic events[1,2]. Hence, platelet inhibition is the cornerstone of primary and secondary prevention in patients with cardiovascular disease[3]. Recently, non-canonical effects of platelets in multiple disease processes were revealed ranging from protective features in tissue regeneration and vascular integrity to detrimental effects in tumor progression and metastasis[4]. There, the hundreds of biomolecules platelets release such as chemokines, growth factors, adhesion molecules, immune mediators, polyphosphatases and coagulation factors have been implied to play important roles[5,6]. Although the outcome of patients with acute myocardial

infarction (AMI) is determined by urgent revascularization and antiplatelet therapy[7], a multitude of processes is involved in the consecutive myocardial response such as myocardial death, local and systemic inflammation, and scar formation[8]. The platelet secretome has been implicated in the orchestration of many of these processes[9]. Indeed, platelets promote inflammation by releasing chemokines (PF4, RANTES, CXCL4, CXCL5, CXCL12, CXCL14), selectins, and integrins (P-selectin, GPIb alpha, ICAM-2, GPIIb/IIIa, CD147, CD40L)[10]. Platelets also trigger cardiac fibroblast activation and myofibroblast transdifferentiation by releasing growth factors and bioactive amines such as serotonin and histamine-major determinants of cardiac scar

[1]Department of Cardiology, Pulmonology, and Vascular Medicine, University Hospital Düsseldorf, Medical Faculty of the Heinrich Heine University Düsseldorf, Düsseldorf, Germany. [2]CARID, Cardiovascular Research Institute Düsseldorf, Medical Faculty and University Hospital, Düsseldorf, Germany. [3]National Heart and Lung Institute, Imperial College London, London, UK. [4]Institute of Molecular Medicine III, University Hospital Düsseldorf, Heinrich Heine University Düsseldorf, Düsseldorf, Germany. [5]Department of Anesthesiology and Intensive Care, University Hospital Jena, Jena, Germany. [6]These authors contributed equally: Amin Polzin, Lisa Dannenberg, Marcel Benkhoff. ✉e-mail: bodo.levkau@med.uni-duesseldorf.de

formation, fibrosis, and dysfunction[11,12]. Taken together, many platelet-derived molecules appear to impair myocardial healing after AMI[13–17]. However, other studies have demonstrated exactly the opposite: a protective and regenerative role of platelets during myocardial healing. In fact, cardioprotective properties have been found for many platelet molecules such as adenine nucleotides[18], thrombospondin-1[19], lipoxin A4[20], maresin 1[21], annexin A1[22], TGF-β1[23], and SDF1-α[24].

Sphingosine-1-phosphate (S1P) is an important bioactive lipid released from platelets upon activation[25] and has potent cardioprotective properties in AMI when acutely administered or kept constitutively high due to pharmacological or genetic inhibition of its degradation[26,27]. In humans, S1P levels in plasma are dynamically regulated during AMI[28–30] or even transient ischemia during percutaneous coronary intervention[31]. In our study, we hypothesized that platelets may be an important source of the S1P burst during AMI and that platelet-derived S1P is relevant for the extent of myocardial damage due to AMI.

Here, we address these questions in experimental AMI in mice lacking components of the S1P synthesis, release, and signaling pathways. We show the mechanisms underlying platelet-associated cardioprotection and finally, reveal the importance of platelet-released S1P on admission in patients with ST-elevation myocardial infarction (STEMI) in terms of infarct size (IS) and long-term prognosis.

## Results

### SNT+ reduces IS and improves cardiac outcomes after AMI
To test whether activation of platelets resulted in the release of factors that protect the heart during AMI, we administered cell-free supernatants of murine platelets either activated by ADP (SNT+) or not (SNT−) intravenously to mice five minutes prior to AMI. SNT+ clearly decreased infarct size by 24% on histomorphometry (Fig. 1a, b) and improved cardiac function on echocardiography by 21% (Fig. 1c), respectively, as compared to SNT− after 24 h. SNT− showed no effect on infarct size and cardiac function as compared to vehicle-treated controls. Neutrophil accumulation in the infarct area (Ly6G+ cells) was 55% lower one day after ischemia (Fig. 1e), whereas apoptosis (active caspase-3 staining) was reduced by 41% five days after ischemia in SNT+ compared to SNT− treated mice (Fig. 1f). The smaller infarct size observed within SNT+ treatment translated into much-improved cardiac systolic function on follow-up after AMI (21 days) as indicated by an 18% higher stroke volume (Fig. 1g) and a 15% higher cardiac output (Fig. 1h). Functional echocardiography data over time can be seen in Supplementary Table 1. Not only ADP-generated but also collagen-generated SNT+ exerted cardioprotection (Supplementary Fig. 1).

### Platelet-dependent cardioprotection depends on Sphk1
One of the many factors released by activated platelets is S1P[9,14,32]. To test whether this was the case in our model and if so, whether the released S1P may have contributed to the cardioprotection by SNT+, we first measured S1P concentrations in SNT+ and SNT−. S1P concentrations were, indeed, 25% higher in SNT+ compared to SNT− (Fig. 1k). To understand its contribution to cardioprotection by SNT+, we performed AMI experiments as explained above, but this time with SNT+ and SNT− derived from sphingosine kinase 1 (Sphk1)-deficient mice and wildtype controls. S1P concentrations were 70% lower in SNT− of SphK1-deficient platelets and failed to increase after activation in SNT+ (Fig. 1k). As a first indication that S1P is involved in cardioprotection by SNT+, the SNT+ derived from SphK1-deficient platelets did not reduce infarct size compared to SNT− (Fig. 1l, m).

### Platelet-derived S1P exerts cardioprotection via S1P_1
To test whether signaling by S1P receptors was causally involved in cardioprotection by platelet-released S1P in SNT+ and if so, whether it acted directly on the heart rather than systemically, we analyzed the effect of SNT+ and SNT− in a Langendorff model of global ischemia,

where cardioprotective effects on local cardiac cells can be assessed without effects on systemic immune cell responses. We did so in the presence of the pharmacological $S1P_1$ inhibitor W146 because of the importance of $S1P_1$ in AMI[33]. Here, SNT+ reduced infarct size by 36% compared to SNT− (Fig. 2a) and improved cardiac recovery as reflected by a 70% increased left ventricular pressure gradient (DP, Fig. 2b) and by a 47% lower left ventricular end-diastolic pressure (LVEDP, Fig. 2c) as compared to SNT−. In the presence of W146, cardioprotection by SNT+ was completely blunted (Fig. 2a–c). We then corroborated these findings by using cardiomyocyte-specific $S1P_1$-deficient mice ($S1P_1^{\alpha MHCCre+}$)[34]. There, the cardioprotective effect of SNT+ was present only in wildtype ($S1P_1^{\alpha MHCCre-}$) but not in $S1P_1^{\alpha MHCCre+}$ Langendorff hearts suggesting a direct and local cardioprotective effect mediated by cardiomyocyte $S1P_1$ (Fig. 2e, f). The other two S1PRs ($S1P_2$ and $S1P_3$) relevant in the heart are not altered in this mouse line[33]. Finally, to extend these finding back to the situation in vivo, we performed AMI in $S1P_1^{\alpha MHCCre-}$ and $S1P_1^{\alpha MHCCre+}$ mice and observed that SNT+ again protected only $S1P_1^{\alpha MHCCre-}$ but not $S1P_1^{\alpha MHCCre+}$ mice (Fig. 2g, h).

### Deficiency of S1P release by Mfsd2b leads to increased IS
Although an increase of plasma S1P during AMI has been well documented[35,36], the origin of the increase has remained less clear. Thus, we measured S1P in plasma, platelets, red blood cells (RBC), and myocardium at different points in time after AMI. Five minutes after ischemia, S1P levels were significantly increased in the peripheral circulation in plasma (Fig. 2i), mirrored by an 80% decrease of platelet S1P in the same blood sample (Fig. 2j). In contrast, two other potential sources (RBC and the heart) did not show any changes (Fig. 2k, l) suggesting that the increase in circulating S1P during AMI may stem from platelet activation.

The S1P transporter *Mfsd2b* has recently been shown to be responsible for S1P release from activated platelets[37,38]. We thus generated SNT+ and SNT− from *Mfsd2b*-deficient mice and observed that SNT+ was unable to decrease infarct size (Fig. 2m, n). Finally, we asked whether endogenous release of platelet-derived S1P during AMI by *Mfsd2b*—a scenario that most closely mimics the situation in patients in vivo—might be cardioprotective. Remarkably, *Mfsd2b*[−/−] mice had a 39% larger infarct size compared to *Mfsd2b*[+/+] after AMI in vivo (Fig. 2o, p). We excluded cardiac cell-specific effects of *Mfsd2b*, as infarct sizes were similar between in isolated *Mfsd2b*[−/−] and *Mfsd2b*[+/+] hearts in the Langendorff model (Fig. 2q).

### GPIIb/IIIa but not P2Y12 antagonism preserves cardioprotection
Inhibition of platelet aggregation is standard of care in patients with AMI. However, drugs that inhibit platelet aggregation may affect S1P release. Indeed, we have previously shown that acetylsalicylic acid (ASA) inhibited S1P release from activated platelets in a cyclooxygenase (COX)-dependent manner[25]. Here, we tested if two other classes of antiplatelet agents with distinct mechanisms of action that are used during AMI—an antagonist of the platelet ADP receptor P2Y12 (cangrelor) and an antagonist of the platelet glycoprotein (GP)IIb/IIIa (tirofiban)—had any effect on S1P release and perhaps on cardioprotection by SNT+. We first measured S1P release in the presence of these two inhibitors and observed that cangrelor, but not tirofiban, inhibited S1P release in SNT+ (Fig. 3a, Supplementary Fig. 2). Furthermore, SNT+ from tirofiban, but not from cangrelor-treated platelets, was cardioprotective in murine AMI in vivo (Fig. 3b, c). Control experiments using light transmission aggregometry showed that cangrelor and tirofiban were similarly effective in inhibiting platelet aggregation (Fig. 3d) and that cangrelor, but not tirofiban, inhibited p-selectin mobilization (Fig. 3e) and ATP release (Fig. 3f).

### S1P concentrations are associated with outcome after STEMI
To investigate if plasma S1P concentrations are relevant to clinical outcome in patients with AMI, we examined the association between

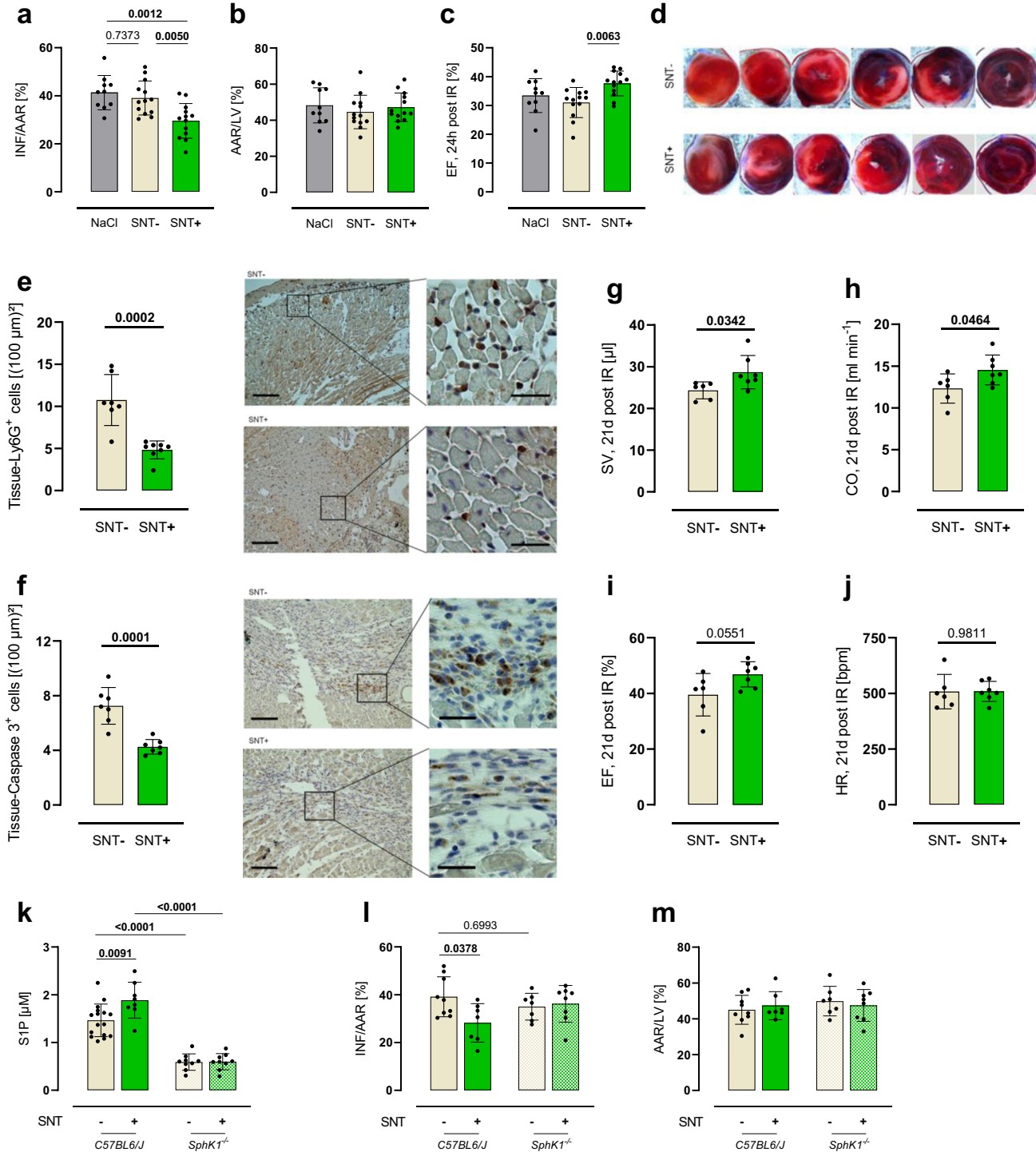

**Fig. 1 | Supernatant of activated platelets reduces infarct size and improves cardiac outcome after myocardial infarction—this depends on platelet sphingosine kinase 1. a** Injection of cell-free supernatant of activated platelets (SNT+) prior AMI leads to decreased infarct size (INF/AAR) after 24 h of reperfusion as compared to supernatant of non-activated platelets (SNT−) and NaCl (ANOVA-analysis, $n_{NaCl} = 10$, $n_{SNT−} = 13$, $n_{SNT+} = 13$). **b** Area at risk per left ventricle (AAR/LV) did not differ between the groups. **c** Echocardiographic assessment 24 h post AMI showed improved cardiac function (EF = ejection fraction) in SNT+ treated mice (ANOVA-analysis, $n_{NaCl} = 10$, $n_{SNT−} = 13$, $n_{SNT+} = 13$). **d** Exemplary images of TTC-stained hearts. **e** This reduction of infarct size by SNT+ was associated with reduced neutrophil accumulation 24 h after AMI (unpaired $t$ test, $n_{SNT−} = 7$, $n_{SNT+} = 8$, scale

bar left: 100 μm, scale bar right: 30 μm). **f** Apoptosis was reduced five days after AMI (unpaired $t$ test, $n_{SNT−} = 7$, $n_{SNT+} = 7$, scale bar left: 100 μm, scale bar right: 30 μm), and **g–j** cardiac function was improved 21 days after AMI (SV = stroke volume, CO = cardiac output, HR = heart rate, unpaired $t$ test, $n_{SNT−} = 6$, $n_{SNT+} = 7$). **k** S1P concentrations in SNT+ was 25% higher as compared to SNT−. S1P concentrations in SNT− of mice lacking $SphK1$ were 70% lower and did not increase in SNT+ (two-way ANOVA, C57BL6/J $n_{SNT−} = 16$, $n_{SNT+} = 8$, $SphK1^{-/-}$ $n_{SNT−} = 9$, $n_{SNT+} = 9$).
**l** Accordingly, SNT+ of $SphK1$-deficient platelets fails to reduce infarct size (two-way ANOVA, C57BL6/J $n_{SNT−} = 9$, $n_{SNT+} = 7$, $SphK1^{-/-}$ $n_{SNT−} = 7$, $n_{SNT+} = 8$). **m** No difference in the area at risk was observed. Error bars in each panel represent standard deviation.

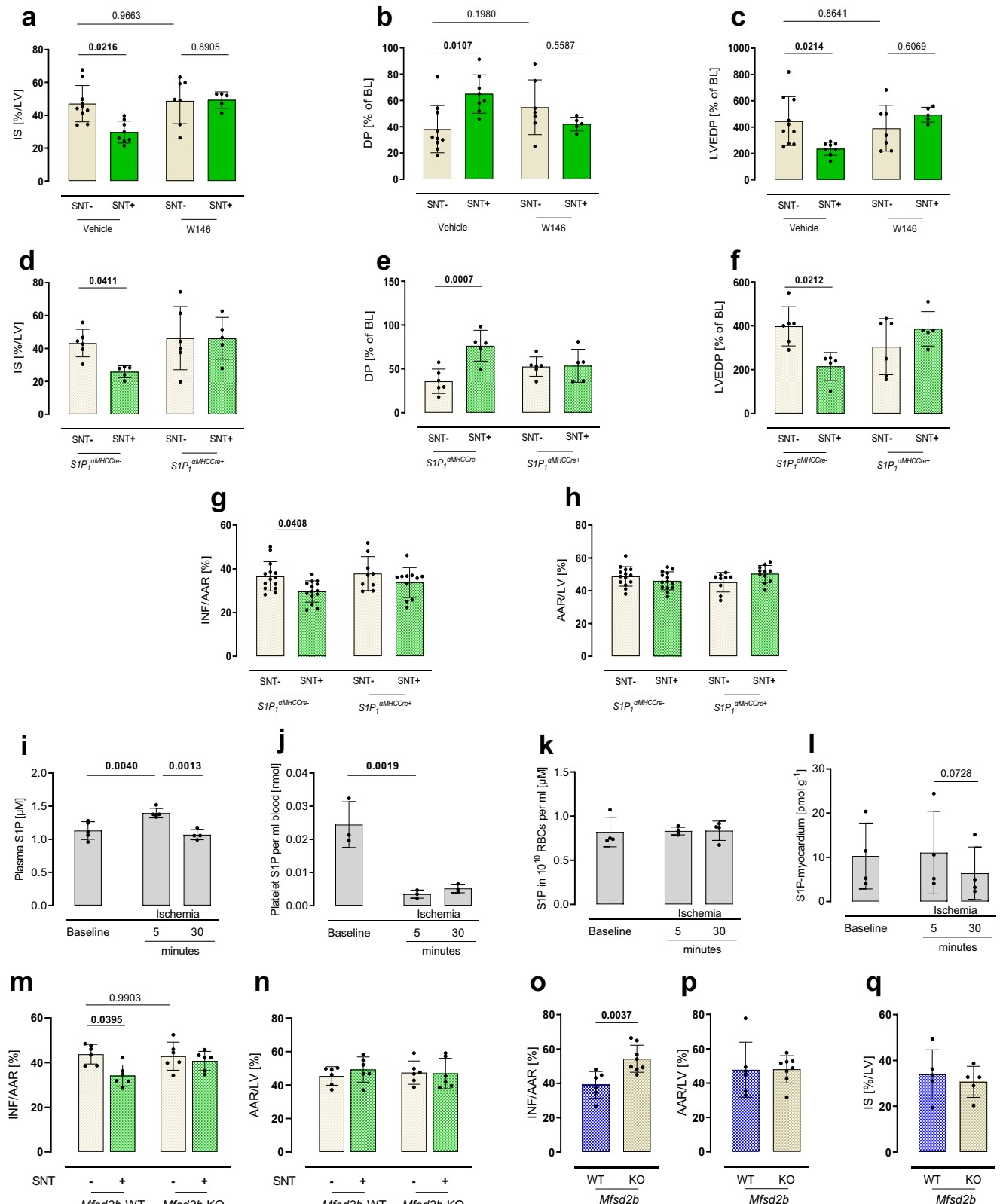

plasma S1P concentration at admission and cardiovascular mortality during the first 12 months in 127 STEMI patients. Mortality as analyzed by plasma S1P terciles clearly showed that higher S1P concentrations were associated with lower cardiovascular mortality (Fig. 3g). Furthermore, plasma S1P levels at admission were negatively correlated with infarct size after 6 months as assessed by MRI (Fig. 3h). For a detailed overview of patients who were not eligible for cardiac

magnetic resonance imaging, we provided a flow chart (Supplementary Table 2). S1P concentrations on day 1 and day 5 were not associated with mortality. Patient characteristics and co-medication showed no differences in any other parameter (Supplementary Tables 3 and 4). We performed a multivariate regression of the infarct size and multivariate analyzes before and after inverse probability treatment weighting as matching analysis (IPTW). High S1P concentrations were a robust

**Fig. 2 | Platelet-derived S1P exerts cardioprotection through the cardiomyocyte S1P receptor 1 (S1P$_1$) and deficiency of platelet S1P transporter major facilitator superfamily domain containing 2B (*Mfsd2b*) leads to increased infarct size in vivo. a** SNT+ leads to decreased infarct size (IS [%LV]), **b** higher left ventricular pressure gradient (DP), and **c** lower left ventricular end-diastolic pressure (LVEDP) in a model of Langendorff-perfused hearts. This was blunted by pharmacological inhibition of S1P receptor 1 via W146 treatment (S1P$_1$; two-way ANOVA, Vehicle $n_{SNT-} = 10$, $n_{SNT+} = 8$, W146 $n_{SNT-} = 7$, $n_{SNT+} = 5$). **d** Accordingly, genetic cardiomyocyte deficiency of S1P$_1$ ($S1P_1^{aMHCCre+}$) abrogated beneficial SNT+ effects in Langendorff-perfused hearts in terms of infarct size, **e** left ventricular pressure gradient (DP) and **f** LVEDP (two-way ANOVA, $S1P_1^{aMHCCre-}$ $n_{SNT-} = 6$, $n_{SNT+} = 5$, $S1P_1^{aMHCCre+}$ $n_{SNT-} = 6$, $n_{SNT+} = 5$). **g** In vivo AMI experiments in these animals revealed preserved cardioprotection in littermate controls but blunted reduction of infarct size (INF/AAR) in cardiomyocyte-specific S1P$_1$ deficiency (Two-way

ANOVA, $S1P_1^{aMHCCre-}$ $n_{SNT-} = 14$, $n_{SNT+} = 14$, $S1P_1^{aMHCCre+}$ $n_{SNT-} = 9$, $n_{SNT+} = 11$). **h** No differences in area at risk (AAR/LV) were observed. **i** Time-series analyses of plasma and circulating cell S1P concentrations in mice undergoing in vivo AMI revealed an increase in plasma S1P (ANOVA-analysis, $n = 4$). **j** This was mirrored by reduced platelet S1P content (ANOVA-analysis, $n = 3$). **k, l** No changes in red blood cells (RBC) and myocardium were observed (ANOVA-analysis, $n = 4$). **m** In addition, SNT+ of *Mfsd2b*-KO mice did not decrease infarct size of treated C57BL/6 J mice (Two-way ANOVA, $n = 6$), **n** while now changes of area at risk were detectable. **o, p** Deficiency of platelet S1P exporter *Mfsd2b* showed increased infarct size during in vivo AMI compared to littermate controls (unpaired *t* test, $n_{WT} = 6$, $n_{KO} = 8$). **q** Cardiac-specific effects in *Mfsd2b*-deficient animals were ruled out by similar infarct sizes in ex vivo Langendorff experiments (unpaired *t* test, $n_{WT} = 5$, $n_{KO} = 5$). Error bars in each panel represent standard deviation.

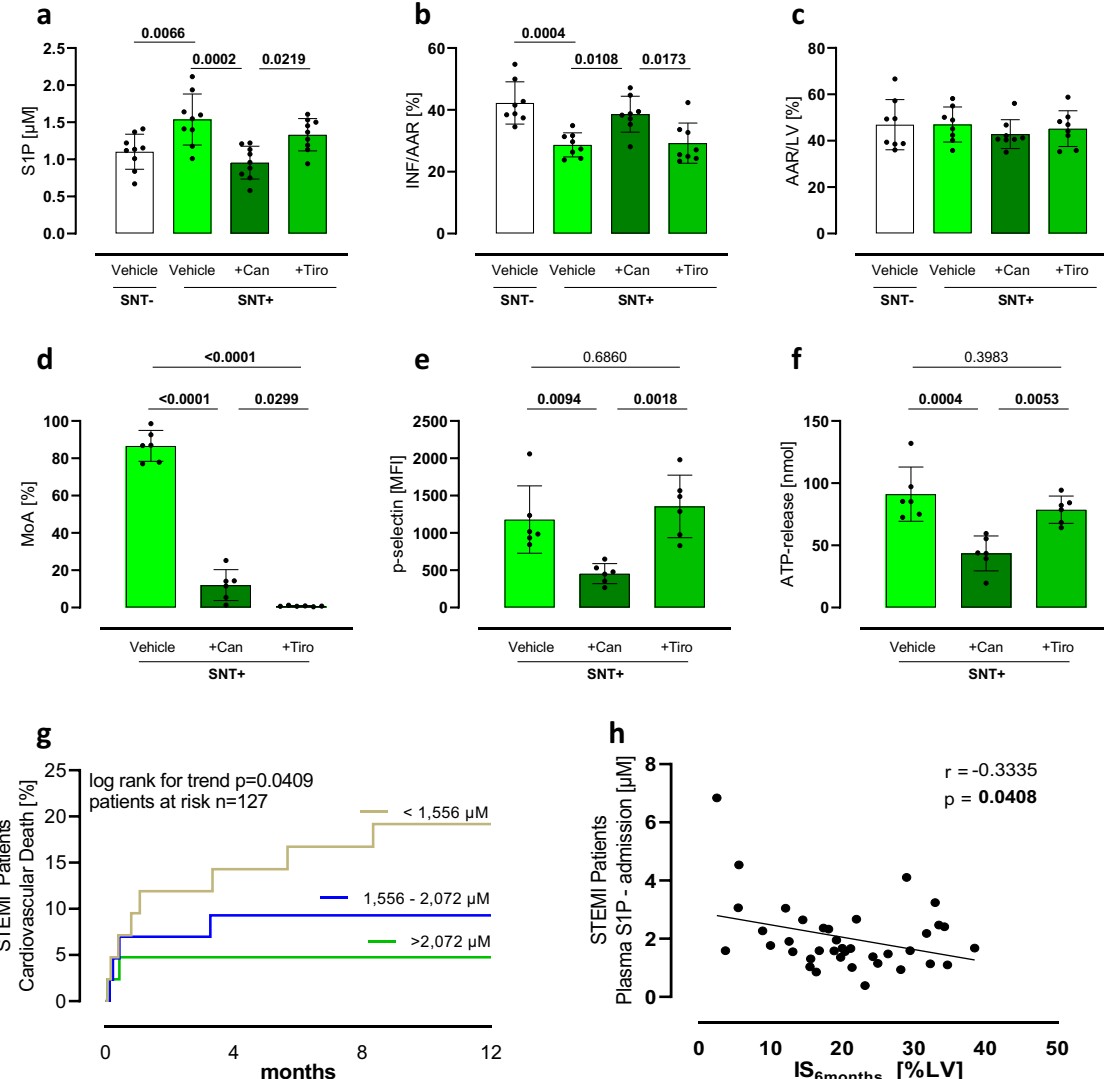

**Fig. 3 | S1P concentrations are associated with outcome of patients with STEMI - inhibition of platelet aggregation by GPIIb/IIIa but not P2Y12 antagonists preserves platelet S1P release and cardioprotection. a** Platelet S1P release during activation was reduced by P2Y12 inhibition (Can=Cangrelor) but preserved during GPIIb/IIIa inhibition (Tiro=Tirofiban, ANOVA-analysis, $n = 9$). **b, c** This translated to blunted cardioprotection by activated platelet supernatant (SNT+) of cangrelor-treated platelets but sustained cardioprotection during tirofiban treatment (ANOVA-analysis, $n = 8$). **d** As expected platelet aggregation was sufficiently

inhibited by both cangrelor and tirofiban (maximum of aggregation (MoA), ANOVA-analysis, $n_{Vehicle} = 6$, $n_{Can} = 6$, $n_{Tiro} = 6$). **e, f** In contrast, *alpha*-granule (**e**) and *dense*-granule (**f**) release were not affected by tirofiban (ANOVA-analysis $n_{Vehicle} = 6$, $n_{Can} = 6$, $n_{Tiro} = 6$). **g** Plasma S1P concentrations were associated with cardiovascular death and **h** infarct size in patients with ST-elevation myocardial infarction (Pearson correlation, $n = 38$). Error bars in each panel represent standard deviation.

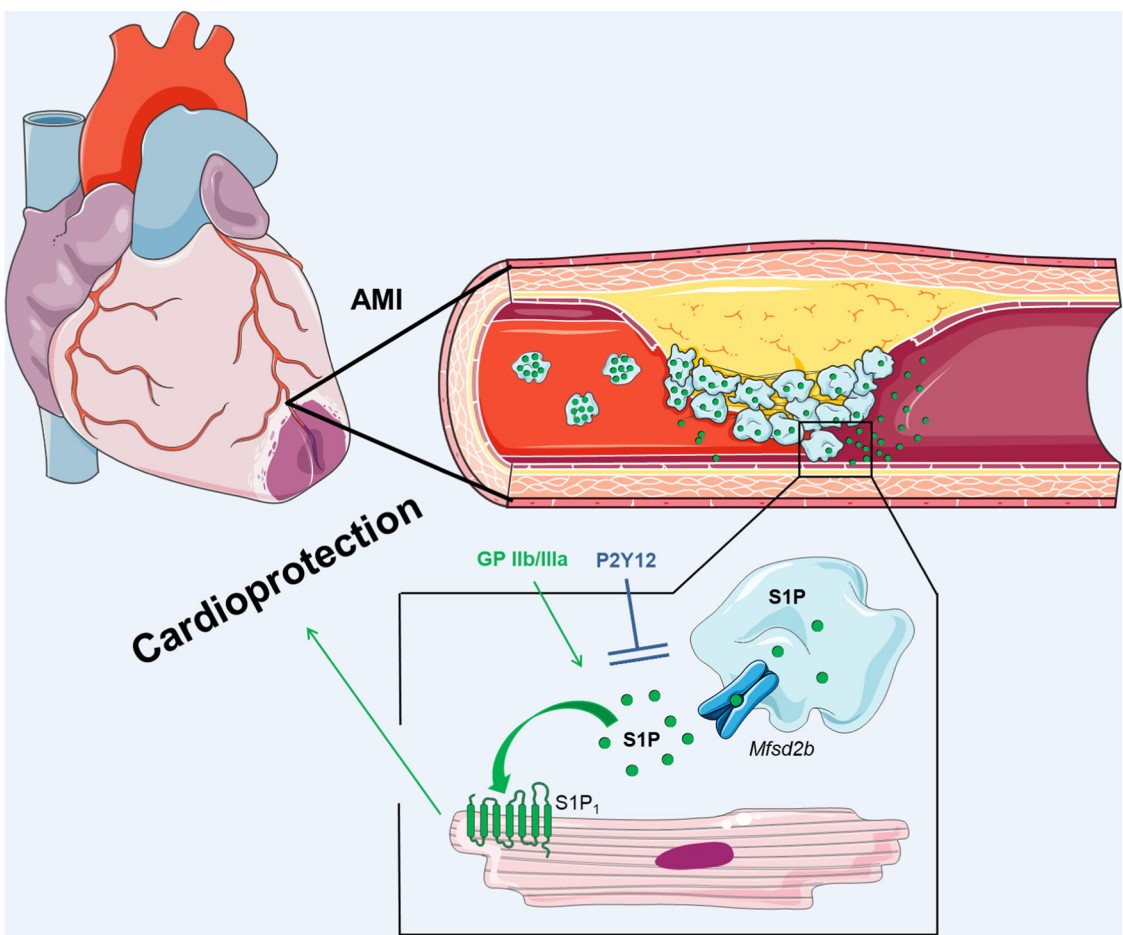

Fig. 4 | Main findings of this study. Activated platelets release S1P during acute myocardial infarction (AMI) through the S1P exporter *Mfsd2b*; platelet-derived S1P during AMI is cardioprotective by engaging the cardiomyocyte $S1P_1$; platelet S1P-mediated cardioprotection is preserved with GPIIb/IIIa antagonists, but not P2Y12 antagonists. The Figure was partly generated using Servier Medical Art, provided by Servier, licensed under a Creative Commons Attribution 3.0 unported license (https://creativecommons.org/licenses/by/3.0/).

predictor of death in our cohort (unweighted: $p = 0.04$, IPTW: $p = 0.03$, Supplementary Table 5). Furthermore, S1P was already able to improve the reclassification of patients when tested over GRACE risk score. ROC analysis of GRACE score including S1P concentrations showed increased AUC compared to the original GRACE score without S1P concentrations (Supplementary Fig. 3).

## Discussion

The main findings of this study are: (I) activated platelets release S1P during AMI through the S1P exporter *Mfsd2b*; (II) platelet-derived S1P during AMI is cardioprotective by engaging the cardiomyocyte $S1P_1$; (III) platelet S1P-mediated cardioprotection is preserved with GPIIb/IIIa antagonists, but not P2Y12 antagonists and (IV) higher plasma S1P concentrations in STEMI patients at admission are favorably associated with infarct size and cardiovascular mortality during 12 months follow-up (Fig. 4).

Platelets are critically involved in the occurrence of ischemic events, and antiplatelet therapy is standard of care in AMI treatment[1,2]. Although such therapy has proven to be a major achievement in cardiovascular medicine, its resounding success may have obscured some of the more discrete but nevertheless important platelet functions. In fact, activated platelets release a plethora of bioactive substances with manifold effects on neighboring cells and tissues that would be missing when platelet activation is inhibited[18–24]. Here, we were fortunate to unveil platelet S1P as a hereto unknown cardioprotective function of the platelet secretome in the setting of AMI. In addition, we were able

to show how important even a small amount of released S1P can be for cardioprotection (Supplementary Fig. 4). The *Mfsd2b* receptor seems to be particularly important here, as the lack of *Mfsd2b* results in 50-60-fold accumulation of S1P in platelets. Mice lacking this receptor showed significantly increased infarct size, due to missing S1P "burst" during AMI. Many reports have shown that administration of exogenous S1P protects against ischemia/ reperfusion (I/R) injury in several organs and that interfering with the generation or degradation of endogenous S1P determines the extent of I/R injury[34,39]. However, the dynamics and sources of circulating plasma S1P that are verifiably present during acute coronary syndrome and murine AMI have been largely unknown. In general, the reasons why certain patients have higher S1P levels than others are unclear. S1P levels vary widely between individuals. Reasons for this are for example ethnicity[40], obesity[41], and body fat percentage[42]. Moreover, age and gender influence S1P. However, data is regarding this is contrary[43–46]. Although many pathological conditions such as cardiovascular and infectious diseases, inflammation, sepsis and metabolic disorders are associated with altered homeostatic S1P concentrations in plasma[26,27,35], only few exhibit acute S1P bursts in plasma as in AMI[47]. Indeed, a time- and ischemia severity-dependent increase of S1P has been described in angina pectoris and AMI[48]. This is in line with a rapid event-associated time-limited burst of S1P release followed by normalization, e.g., due to the short half-life of S1P in plasma[31]. Nevertheless, such transient S1P bursts are not an AMI epiphenomenon, but clearly have a biologically relevant meaning in STEMI patients: the height of the S1P burst was

positively associated with survival and negatively with infarct size. Interestingly, only S1P concentrations at admission determined outcome. However, S1P levels on day 1 or day 5 did not. The reason for this is unclear but might be due to medication in STEMI patients. Especially, as we could show that P2Y12 inhibition interferes with S1P release from platelets. This needs to be explored in a prospective setting, but if successful, it may define a novel prognostic marker.

The source of AMI-associated S1P bursts had not been identified until now. Here, we show that activated platelets are the main source in contrast to other hematopoietic cells or the heart itself. Thrombin activation of platelets during clotting has been demonstrated to cause S1P release in a COX-dependent manner[25,49]. Vice versa, S1P synergizes with thrombin to induce the release of tissue factor from platelets[50,51]. This has led to the ostensible notion that S1P may either be a promiscuous marker of platelet activation or, assuming causal participation, even deleterious in AMI. Our data suggest exactly the contrary: platelet-released S1P is beneficial during AMI by limiting myocardial damage. Unsurprisingly, this effect has been difficult to identify so far as it is concealed beneath the widely implemented antiplatelet therapy in AMI and the valid therapeutic paradigm behind it.

However, our study reveals an exploitable therapeutic window as we have observed that not all antiplatelet drugs prevented S1P release to the same extent. In AMI patients, dual antiplatelet therapy with ASA and a P2Y12 inhibitor is currently recommended[52,53]. While the P2Y12 inhibitor cangrelor blocked platelet S1P release and blunted cardioprotection by activated platelet supernatant, the GPIIb/IIIa inhibitor tirofiban preserved both S1P release and cardioprotection, whereas both drugs achieved efficient anti-aggregation. Hence, an important conclusion of our study is that an antiplatelet strategy that conserves platelet S1P release during AMI would be the most desirable one. In the clinical setting, GPIIb/IIIa inhibitors were shown to reduce thrombotic or ischemic complications post PCI[54,55] and the incidence of major adverse cerebro- and cardiovascular events (MACCE)[56,57]. Furthermore, GPIIb/IIIa inhibitors are known to improve ST-resolution[58] and reduce infarct size[59] in the setting of AMI. However, they are currently only recommended for: (a) up-stream therapy in high-risk patients with high thrombus burden, (b) thrombotic complications[60], and (c) bail-out in slow or no-reflow conditions due to thrombus formation[52,53,61]. The reasons for this "second line" recommendation are primarily that, (a) they enhance the risk of bleeding (like any other antithrombotic substance)[57,62,63] and (b) were not tested in presence of more potent P2Y12 inhibitors like prasugrel, ticagrelor or cangrelor. However, we have learned much since the GPIIb/IIIa inhibitors landmark trials[54,55]. Especially peri-procedural bleeding was diminished substantially[56,57] due to reduction of access site bleeding by choosing radial access[64]. Additionally, P2Y12 medication prior to decision of percutaneous coronary intervention in AMI is not recommended, and deescalation of antithrombotic treatment early after AMI reduced the risk of bleeding without enhancing ischemic events[52,65]. Certainly, much more has to be examined and pharmacologically pursued, if the paradigm we have defined here—that of cardioprotection by platelet-derived bioactive molecules such as S1P—matures and becomes clinically relevant. If and how GPIIb/IIIa antagonists could be employed to exploit the benefits of such concealed cardioprotection awaits clinical proof. However, a possible scenario that can be implemented right away with the current treatment options in order to investigate it could be, e.g., hard and early platelet inhibition by GPIIb/IIIa inhibition to tackle ischemic events followed by deescalated P2Y12 inhibition or single antiplatelet therapy. We believe that the recognition of this platelet-mediated intrinsic protection against tissue injury should be taken into consideration in the treatment of many settings where antiplatelet therapy is indicated but could be modified in an attempt to preserve and exploits its benefits. This may be an exciting therapeutic paradigm in platelet medicine that reaches far beyond the reduction of infarct size in AMI.

## Methods

### Animals
Wildtype mice were purchased from Janvier Labs (Saint-Berthevin, France). Edg1 Cardio Cre mice ($S1P_1^{\alpha MHCCre+}$ /$S1P_1^{\alpha MHCCre-}$) were bred as previously published[33]. SphK1- deficient mice were gifted by Richard Proia, NIH and *Mfsd2b* knockout mice were obtained from the MMRRC at UCDavis. Mice at the age of 12–15 weeks at the beginning of experiments were used. All animals had free access to drinking water and standard diet ad libitum.

### Generation of activated and non-activated supernatant (SNT)
Blood was taken from donor mice via retrobulbar plexus with a heparinized cannula. Afterwards, heparinized whole blood rested for 15 min at 37 °C. After first centrifugation (170 g, 5 min, room temperature (RT)) platelet rich plasma (PRP) was taken and incubated at 37 °C with ADP (final concentration 10 μM) or collagen (10 μg/ml, Probe & go Labor Labordiagnostica GmbH, Germany). After 3 min of incubation SNT+ was generated via second centrifugation (2000g, 5 min, RT). SNT− was incubated with the same volume of saline solution. If required, inhibitors were added after first centrifugation and 2 min prior to ADP incubation. They were used with following final concentrations: 200 nM cangrelor (Can, 200 μM, Sigma Aldrich, St. Louis, Missouri), 50 nM tirofiban (Tiro, 50 nM, Aggrastat®, Correvio).

### Determination of S1P concentrations
Lipid extraction and liquid chromatography-mass spectrometry (LCMS; Merck–Hitachi Elite LaChrom System) were performed to measure S1P levels in plasma[66].

Platelets were resuspended in Methanol (MeOH), followed by addition of 10 μL internal standards (for S1P (d18:1): 10 pmol C17 S1P in MeOH; for S1P(d17:1): d7 S1P in MeOH, Avanti Polar Lipids Inc., Alabaster, AL). Samples were mixed, precipitated overnight at −80 °C and centrifugated (5 min at 21,300 × g, 4 °C). The supernatant was transferred into mass spectrometry sample vials and stored at −80 °C until measurement.

Tissue samples (10–20 mg were homogenized in a Stomacher Bag with 10 μL internal standard solution and 500 μL MeOH. The homogenate was collected, another aliquot MeOH (500 μL) was added to the stomacher bag and, after rinsing, the solution was pooled with the first homogenate (total 1 mL). Samples were mixed, precipitated overnight at −80 °C, and centrifuged (5 min at 21,300 × g, 4 °C). The supernatant was transferred into mass spectrometry sample vials and stored at −80 °C until measurement.

Chromatographic separation was performed on a LCMS-8050 triple-quadrupole mass spectrometer (Shimadzu Duisburg, Germany) interfaced with a Dual Ion Source and a Nexera X3 Front-End-System (Shimadzu Duisburg, Germany). HPLC is performed with a 2 × 60 mm MultoHigh-C18 RP column with 3 μm particle size at 40 °C. Mobile phases consisted of [A] 1% aq. HCO2H and [B] MeOH. The following LC gradient was used: Start at 10% [B], linear increase of [B] from 10% to 100% from 0 min to 3.0 min (B-Curve = −2), 100% [B] from 3.00 min to 8 min and equilibration from 8.01 min to 10.00 min prior next injection. Flow rate was 400 μL/min and injection volume of all samples was 10 μL. MS settings were the following: Interface: ESI, Nebulizing Gas flow: 2 L/min, Heating Gas Flow: 10 L/min, Interface Temperature: 300 °C, DL Temperature: 250 °C, Heat Block Temperature: 400 °C, Drying Gas Flow: 10 L/min. Data were collected using multiple reaction monitoring (MRM). Positive ionization was used for qualitative analysis and quantification. Standard curves were generated by measuring increased amounts (100 fmol to 50 pmol) of S1P (d18:1) and C17 S1P (d17:1) together with internal standard (C17 S1P (d17:1) for S1P (d18:1); d7 S1P (d18:1) for C17 S1P (d17:1), 0.3 μM final conc. in MeOH). Linearity of the standard curves and correlation coefficients were obtained by linear regression analysis. All MS analyses were performed using LabSolutions 5.99 SP2 and LabSolutions Insight (Shimadzu Duisburg, Germany).

## Isolated mouse heart perfusion

Langendorff experiments were conducted to study the cardioprotective effects on local cardiac cells without influence of circulating blood cells[67]. In brief, Krebs-Henseleit Buffer (KHB) including 118 mM NaCl, 4.7 mM KCl, 0.8 mM $MgSO_4$, 25 mM $NaHCO_3$, 1.2 mM $KH_2PO_4$, 5 mM glucose, 110 mM Na-pyruvate and 2.5 mM $CaCl_2$ was warmed at 37 °C and gassed with carbogen circulating in the Langendorff- apparatus (Hugo Sachs, March-Hugstetten, Germany). Wildtype mice (C57BL/6 J) and Edg1 CardioCre+/− mice ($S1P_1^{aMHCCre+}$ /$S1P_1^{aMHCCre}$) were used for experiments. DMSO and $S1P_1$ antibody W146 were applied 30 min prior to experiment. Mice were anesthetized by intraperitoneal (i.p.) injection of ketamine (100 mg/kg; Pfizer Pharma PFE GmbH, Berlin, Germany) and xylazine (10 mg/kg; Bayer, Leverkusen, Germany) followed by i.p. injection of heparin (1000 IU/mouse; Rotexmedica, Trittau, Germany). After loss of reflexes (5–7 min after injection), hearts were extracted, prepared rapidly, and placed in 4 °C cold KHB. The ascending aorta was cannulated and fixed by a node. Hearts were placed into the Langendorff system and retrogradely perfused with a constant KHB flow measured by ultrasound flow probe (Type MC1PRB-HSE for HSE-TTFM; Hugo Sachs, March-Hugstetten, Germany) at 100 mmHg. A small water-filled balloon adapted to a pressure transducer was inserted in the left ventricle (LV) for quantification of LV pressure gradient (DP) and for LV end-diastolic pressure (LVEDP). Hearts were paced at 600 beats per minute. After equilibration of 20 min, 20 seconds test ischemia was conducted to investigate flow reserve. Hearts that did not reach a minimum of 70% flow increase were excluded from further analyses. Afterwards, hearts recovered for further 5 min. Baseline parameters were assessed. Hearts that did not reach a minimum of 50 mmHg DP were excluded as well. Afterwards, global flow was interrupted and preloading with murine platelet supernatant with and without prior activation (SNT− and SNT+) was conducted via a syringe driver (0.4 ml/min; 400 µl) followed by 40 min of global ischemia. Hearts were warmed and kept moist by a small KHB-containing basin surrounding the ischemic heart. Next, the flow was restarted for two hours of reperfusion with KHB. Hemodynamic parameters were assessed after 60 min. After 120 min, hearts were extracted from the cannula and frozen for one hour at −20 °C. The frozen heart was cut into six slices and 2,3,5-triphenyltetrazolium chloride (TTC, Sigma Aldrich, St. Louis, Missouri) staining was conducted. Infarct size (IS) was calculated as percentage of the left ventricle volume.

## Murine model of acute myocardial infarction (AMI)

Mice of different genotypes underwent ischemia/reperfusion surgery as a murine model of AMI. After intubation of the mice, surgery was conducted under 2 vol% isoflurane anesthesia. AMI was induced via transient left anterior descending artery (LAD) ligation for 30 min[67,68]. Typical ST-elevations were used for monitoring of ligation and resolution. Body temperature was controlled continuously. SNT− or SNT+ respectively was injected intravenously 5 min prior to ischemia. This open-chest surgery was followed by 24 h, 5 days or 21 days of reperfusion. Afterwards, mice were sacrificed and organs were extracted for further analyses.

## Euthanasia of mice

First, animals were anesthetized with ketamine (100 mg/kg; Pfizer Pharma PFE GmbH, Berlin, Germany) and xylazine (10 mg/kg; Bayer, Leverkusen, Germany) by intraperitoneal injection. After loss of reflexes, the thorax was opened and the heart was removed for further analysis.

## Myocardial infarct size measurement

Infarct size was measured 24 h after AMI. Hearts were taken out of the sacrificed mice and surrounding tissue was removed in cold saline solution. Hearts were reperfused with cold saline solution via aorta to remove blood. Ligation of LAD was repeated in a permanent manner.

Afterwards, the hearts were perfused with 1% Evans blue dye (Sigma Aldrich, St. Louis, Missouri), which separates non-ischemic myocardium from area at risk (AAR). Hearts were frozen at −20 °C overnight and then cut into six 1 mm slices starting apical. The slices were incubated for 5 min with 1% TTC solution, which colors vital tissue. Infarct size/AAR-ratio and AAR/left ventricle (LV)-ratio was determined via computer-assisted planimetry.

## Cardiac function measurement by echocardiography

Cardiac function was measured before as well as 24 h and 21 days after AMI by echocardiography[69]. Mice were anesthetized with isoflurane and body temperature was kept at 37–38 °C. The heart rate was monitored during examination. High resolution 18–38 MHz ultrasound transducer (MS 400, VEVO 3100, VisualSonicsInc., Toronto, Canada) was used. To determine left ventricular function parameters, different angles of the heart were measured. Post-processing analyses were performed offline using commercial software (VevoLab 3.2.6., VisualSonicsInc., Toronto, Canada)

## Immunohistochemical staining and quantification

Mice were sacrificed 24 h or 5 days after AMI and perfused with cold PBS. The hearts were removed and fixed in 4% paraformaldehyde (PFA, Alfa Aesar, Massachusetts, USA) overnight. Hearts were then dehydrated and embedded in paraffin. Hearts were cut in 5 µm sections using a microtome (Leica CM 3050 S, Wetzlar, Germany). The heart sections were stained for inflammation and apoptosis. Afterwards, they were deparaffinized and rehydrated and a heat-induced epitope retrieval (HIER) was done using citrate buffer (pH 6.0, ThermoFisher Massachusetts, USA). After the retrieval the sections were blocked with BloxAll-Blocking solution (Vector laboratories, Burlingame, USA). They were then blocked with 2.5 % goat serum and subsequently incubated with primary antibodies at 4°C overnight. The following primary antibodies were used: rat anti Ly6G (1:100, Abcam, Cambridge, UK) and rabbit anti cleaved Caspase-3 (1:400, Cell signaling, Massachusetts, USA). After that, the ImmPRESS®-Goat Anti-Rat IgG (Mouse Adsorbed) Polymer Kit (MP-7444, Vector Labs, Burlingame, Canada) for Ly6G staining and the ImmPRESS®-AP Goat Anti-Rabbit IgG Polymer Kit (MP-7451, Vector Labs, Burlingame, Canada) for Caspase-3 staining were used. The staining was visualized using the ImmPact DAB® (SK-4105, Vector Labs, Burlingame, Canada) detection system. Sections were counterstained with hematoxylin Gill II (T864.1, Roth, Karlsruhe, Germany) and mounted with permanent mounting medium (H-5000, Vector laboratories, Burlingame, Canada). Images of the stained sections were done by using a Leica bright field microscope (DM4000M) with twentyfold magnification. Five fields of 100 µm x 100 µm each were imaged of one heart section and analyzed using ImageJ software.

## Patients and study design

We conducted a hypothesis-generating, prospective, monocentric, time-series, translational analysis in 127 patients with ST-elevation myocardial infarction (STEMI). An all-comers design was applied. Inclusion criteria were age ≥18 years, occurrence of one of the above-mentioned events, and written informed consent. Exclusion criteria were malignant comorbidities and coagulopathies. Written informed consent was obtained from all participants. Blood sampling was conducted during ischemia as well as 12 h, and 5 days after intervention (Several important procedural details (culprit vessel, contrast medium, TIMI flow) can be found in Supplementary Table 5). To assess infarct size, gadolinium-based contrast agent (Gadovist, Bayer Healthcare, Berlin, Germany, 0.2 mmol/kg) was used to detect late gadolinium enhancement in cardiac magnet resonance imaging 6 months after STEMI during clinical follow-up. Post-processing analyses were performed offline using commercial software (cmr42, Circle Cardiovascular Imaging Inc., Calgary, Alberta, Canada and

Extended Workspace, Philips Healthcare, Hamburg, Germany). Infarct size was determined in percent from left ventricle volume. Evaluation of CMR was performed by an experienced cardiologist regarding CMR-imaging. Cardiovascular death was assessed during 12-month follow-up. After sampling, blood was centrifuged at $500 \times g$ for 10 min to gain plasma.

### Light transmission aggregometry (LTA)
Blood from healthy individuals was obtained after written informed consent. First, centrifugation of citrate-anticoagulated whole blood was performed at 1200 rpm for 10 min to generate platelet-rich plasma (PRP). Second, PRP was centrifuged at 13,200 rpm for 5 min to generate platelet-poor plasma (PPP) for calibration of the aggregometer (APACT 4004 LABiTec®). After 2 min of equilibration, induction of platelet aggregation was performed using adenosine diphosphate (ADP, 5 μM Sigma Aldrich, St. Louis, Missouri). If necessary, PRP was incubated with cangrelor (Can, 200 μM, Sigma Aldrich, St. Louis, Missouri) or tirofiban (Tiro, 50 nM, Aggrastat®, Correvio) in time of equilibration. Results were given as maximum aggregation (MoA) in percent.

### Adenosine triphosphate (ATP) release assay
For quantitative analysis of ATP release after platelet activation, bio-luminescence analysis of ATP was performed[70]. Briefly, an ATP Assay Kit (FLAA, Sigma Aldrich, St. Louis, Missouri) was used to determine concentrations of platelet-released ATP. An ATP calibration curve was performed on the same day as the related samples. Washed platelets were incubated with cangrelor (Can, 200 μM, Sigma Aldrich, St. Louis, Missouri) or tirofiban (Tiro, 50 nM, Aggrastat®, Correvio) prior to activation via ADP (5 μM Sigma Aldrich, St. Louis, Missouri).

### Determination of p-selectin expression via flow cytometry
Citrate-anticoagulated whole blood was centrifuged at $300 \times g$ for 10 min to gain PRP. 100 μl of PRP were incubated for 6 min at 37 °C. After another 6 min of stimulation with ADP (5 μM) at 37 °C, the following antibodies were added and incubated for 30 min at 37 °C: CD41/61-APC-Cy7 (Miltenyi Biotec, Bergisch Gladbach, Germany), CD42b-Pe-Cy7 (Invitrogen, Massachusetts, USA) and CD62P-BV421 (BD Biosciences, Franklin Lakes, USA). Before measurement with BD Facs-Verse® (BD Biosciences, Franklin Lakes, USA), the solution was diluted 1:100 in phosphate-buffered saline (PBS). Data were processed and analyzed using Flow-Jo® (BD Biosciences, Franklin Lakes, USA)

### Statistical analyses
For statistical analyses, IBM SPSS©-Software (New York, USA) and GraphPad-Prism© statistical software (GraphPad software Inc, San Diego) were used. Normality of distribution was tested with Kolmogorov–Smirnov test, D'Agostino-Pearson test, qq-plots, and histograms. Normally distributed continues variables were analyzed using $t$ test; non-normally distributed variables using Mann–Whitney $U$ test. Homogeneity of variance was tested with Levene test. In case of heteroscedastic data, Welch test was performed. Comparison of three or more groups was performed using ANOVA-analyses. If the assumption of homogeneity of variances was not met, indicated by a significant Levene test, a Kruskal–Wallis test was performed. Tukey correction was applied in multiple testing. Log-rank test for trend was implemented to test, whether different S1P concentrations are associated with incidence of cardiovascular mortality over time. Additionally, Pearson correlation was used to investigate a possible relation between S1P levels and infarct size. Normality of distribution of variables used for Pearson correlation was assessed with Kolmogorov–Smirnov test. In case of not normally distributed variables Spearman correlation was performed. $P$ values <0.05 were considered significant.

To explore whether S1P concentration after myocardial infarction could improve the predictive power of GRACE score, we first computed the GRACE score for each patient in our sample. We then allocated values to each of the three S1P concentrations. We started with the values 0, 1, and 2 for high, medium and low S1P concentration, respectively, and added them to the GRACE score of patients. We then computed an ROC curve to explore whether the AUC improved. We used Python programing to try out each combination of values for high concentration in the range 0–20, medium concentration in the range 1–40 and low concentration in the range 2–60. Each combination was added to the patients GRACE score and an ROC curve was computed. The combination of values yielding the highest AUC in the ROC analysis was selected.

### Ethics committee approval
The study conformed to the Declaration of Helsinki and was approved by the University of Düsseldorf Ethics Committee. It is registered under clinicaltrails.org with the ID "NCT03539133".

All mice experiments were approved by *Landesamt für Natur, Umwelt und Verbraucherschutz Nordrhein-Westfalen (LANUV)* and in accordance with the European Convention for the Protection of Vertebrate Animals used for Experimental and other Scientific Purposes (Council of Europe Treaty Series No. 123) and 2010/63/EU.

### Reporting summary
Further information on research design is available in the Nature Portfolio Reporting Summary linked to this article.

## Data availability
The data supporting this study can be found in the figures and supplementary information. Source data are provided with this paper.

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

## Acknowledgements

The authors thank Stefanie Becher for experimental support. This work was supported by the Deutsche Forschungsgemeinschaft (DFG, German Research Foundation) Grant No. 236177352 - SFB 1116, TP B11 to B.L. and A.P., TP B6 and B12 to M.K. and by the German Research Foundation (LE 940/7-1 to B.L. and PO 2247/2-1 to A.P). We acknowledge the support of the Susanne-Bunnenberg-Stiftung at the Düsseldorf Heart Center.

## Author contributions

A.P., L.D. M.Be., and B.L. designed the study, analyzed and interpreted data, and wrote the manuscript with the input of M.K. D.M., P.W., M.C., and F.B. performed the human studies and statistical analysis. M.Ba, C.H., P.M, S.A., L.W., J.Z., D.Z., H.H., L.B., N.S., P.K., S.W., D.D., S.S., T.M., M.G., and T.Z. collected and analyzed data and revised the manuscript.

## Funding

## Competing interests

The authors declare no competing interests.
