## [Peer Review File · Nature Communications]

Revealing concealed cardioprotection by platelet Mfsd2b-released S1P in human and murine myocardial infarctionREVIEWER COMMENTS

Reviewer #1 (Remarks to the Author):

The manuscript by Polzin et al. sets out to define a novel cardioprotective role of platelet-derived sphingosine-1-phosphate (S1P) burst in acute myocardial infarction (AMI). The authors show that administration of supernatants from activated platelets reduced infarct size in a mouse model of AMI, which was not observed with platelets deficient for S1P export (Mfsd2B) or production (Sphk1) or in mice deficient for cardiomyocyte S1P receptor 1. Also in STEMI patients, S1P levels appeared to be inversely associated with cardiovascular mortality and infarct size.

The authors employ a straightforward approach to test the cardioprotective benefits of activated platelet supernatant on experimental MI in two different mouse models: (1) in vivo 30 min ischemia followed by 24h, 5 days or 21 days of reperfusion; (2) ex vivo retrograde cardiac perfusion (Langendorff model) and 40 min ischemia followed by 120 min reperfusion. Overall, the study proposes an interesting and innovative concept of translational relevance but currently several major concerns limit overall enthusiasm and would need to be addressed in order to corroborate the conclusions and to increase the impact of the findings.

Major comments:

1. The reduction of the infarcted area in Fig. 1A and Fig. 2C is moderate but significant. It is unclear whether the study and the number of mice used was subjected to a stringent power calculation, given the biological variation associated with the model. This should be critically evaluated to put the effectiveness of a single protective agent in context, but also to confidently assess the underlying mechanisms as postulated. Accordingly, this may help to unravel a significant effect on EF in Fig. 1A.
2. In general, the group size n=4-5 for histology and echocardiography (Fig. 1A-C) is quite low, given the relatively small difference between the two experimental groups. As outlined above, this reviewer would hence recommend a higher n number to increase the data robustness. It is recommended to report functional echocardiography data over time, preferably including sham controls. Alternatively, the baseline and 24h echocardiography data could be provided in a Supplemental table.
3. With regards to the histology data in Fig. 1B, some overview images of cardiac cross-sections should be added to indicate the area of analysis for neutrophils and caspase 3. The time point at day 5 is beyond the neutrophil peak in the infarcted re-perfused heart, which occurs around 24 h post-ischemia.
4. Interestingly, there is a trend towards smaller MI when using Sphk1^{-/-} supernatants (Fig. 1D) even when not activated. Could this be interrogated for a reduction in other, possibly exacerbating factors or is this due to biological variation only? In general, the effects of Sphk1 deficiency are not convincing.
5. Again, in Fig. 2C only small effects are observed and the number of mice per group with n=4 is not sufficient. Labelling is incomplete. Other cell types including RBCs should be shown in Fig. 2D.
6. The exacerbating effect in mice lacking the platelet S1P exporter in Fig. 2E appears more pronounced than the protective effects with a lack of production in Fig. 1E. Could this be due to effects of the transporter on other (lipid) mediators?
7. The manuscript would benefit from a better explanation why two different models were used, e.g. the Langendorff model may allow studying the cardioprotective effects without systemic immune cell responses. Indeed, the cardiomyocyte-specific S1PR KO model supports that the platelet-S1P effect on cardiomyocytes is involved in the reduction of infarct size. This might be further strengthened by co-staining of caspase 3 with cardiomyocytes in the infarct area. Currently, the resolution of the representative immunohistochemical images provided does not allow the reader to distinguish

between positive and negative cells.

8. In Fig. 3A,B, the SNT- control is missing and there does not appear to be a significant difference between the cangrelor and tirofiban groups. Even though there is a trend, it can be questioned whether this would fully explain the effects in Fig. 3B, when taking into account the effects observed with a 70% reduction in S1P observed in Fig. 1D,E. How do the authors explain the high number of replicates in the vehicle group (n=15) as compared to the treatment groups in Fig. 3A, and to the experiments in Fig. 3B.

9. One limitation of the study is that only a single platelet agonist i.e. ADP as was studied. The conclusions drawn, e.g. that inhibiting P2Y12 blocks the release of S1P from platelets may not be valid for other platelet activators contained in ruptured plaques such as collagen. This possibility should at least be acknowledged.

10. Clinical data: the authors have evaluated a prospective cohort of 127 patients with acute STEMI with a 12-month follow-up for hard outcomes (cardiovascular death). However, some aspects are unclear and may influence the interpretation of the data, namely:

a. It is puzzling that the negative correlation in Fig. 3F appears to be largely driven by one patient.

More importantly, can the authors exclude that the effect shown in Fig. 3E can be preserved in a multi-parametric analysis including lipid levels, as S1P may be a surrogate marker for HDL levels.

b. Blood sampling was performed on admission, and at 12 hours and 5 days after the intervention. However, data presented in Fig. 3E,F only refer to S1P concentration on admission. Do the authors observe time-dependent changes in plasma S1P (as in the mouse experiments)? Would S1P levels assessed after intervention still be predictive of mortality?

c. Plasma S1P was evaluated at hospitalization for Fig. 3E,F and the authors state that "blood sampling was performed during ischemia" (lines 365-366). However, there is no mentioning of the interval between the occurrence of symptoms and the sampling (and the intervention). This is crucial, as the authors show the extremely fast and transient release of S1P from platelets in their mouse experiments (Fig. 2D) and as longer "symptom-to-balloon" may account for a worse prognosis in STEMI patients.

d. The nature of the intervention (line 366) is unclear. This reviewer assumes that most (if not all) of the patients underwent primary PCI, but the number of affected/treated lesions, the use of DES or BMS, and the TIMI grade for revascularization are not specified. All these parameters may have a prognostic role in patients with AMI and should therefore be considered.

e. Fig. 3E only includes data for n=38 patients. The authors should report data for all patients in the cohort or explain why a considerable number of cases have been excluded from this analysis.

f. The analyses in Fig. 3E,F are univariate, however, the authors may consider multivariate analyses to adjust the results for other prognostically relevant factors. Indeed, the lack of statistical significance for parameters in tables S1 and S2 may reflect an overall limited statistical power of the investigated cohort. In this regard, the methods paragraph does not report an a priori evaluation of the statistical power, which – ideally - should be provided.

g. It would be interesting to learn whether S1P may improve the reclassification of patients when tested over established risk algorithms (e.g. TIMI, GRACE risk score).

Minor comments:

1. Please add representative images of TTC staining in Fig. 1A. The analysis of apoptosis by staining for active caspase-3 in Fig. 1B does not reveal the cellular context and should be repeated by double immunofluorescence staining including cell-specific markers to distinguish resident from inflammatory cell types. In general, scale bars are missing.

2. Was the observational study registered in any publicly available registry (e.g. ClinicalTrial.org)? If so, the registration ID should be provided.

3. Statistical analyses: (a) it is unclear whether homogeneity of variances has been tested, especially prior to ANOVA and how the authors dealt with heteroscedastic data; (b) it is unclear whether the

authors have assessed normal distribution also before applying bivariate correlation tests; (c) "Tuckey" should rather read "Tukey".

4. In supplemental table 2, it is somewhat surprising that apparently none of the patients was pretreated with a P2Y₁₂ inhibitor and therefore this has not listed. For completeness and because this plays an important role in this paper, this should nevertheless be stated.

Reviewer #2 (Remarks to the Author):

The manuscript 'Revealing concealed cardioprotection by platelet Mfsd2b-released S1P in human and murine myocardial infarction' submitted by Polzin and colleagues indicate that activated platelets release sphingosine 1-phosphate during acute myocardial infarction, which is cardioprotective via stimulation the S1P₁ receptor subtype.

The authors indicate that S1P is released from platelets after activation via the S1P exporter Mfsd2b and binds to the S1P₁ receptor subtype, which is protective against ischemia injury. It is well known from several studies that administration of exogenous S1P protects against ischemia/reperfusion injury. Further studies indicate that S1P is released after thrombocyte activation via the Mfsd2b transporter. Thus, the novelty of the study is an platelet-mediated intrinsic cardioprotection. There are some points in the manuscript that do not seem logical and would need to be addressed. The authors used murine platelet supernatant with and without prior activation via ADP (SNT- and SNT+) to examine the effect on myocardial ischemia. They were able to show that SNT+ contained 25 % more S1P compared to SPT- and that SPT+ was able to decrease infarct size/AAR-ratio measured via computer-assisted planimetry. This experiment should demonstrate the S1P-mediated effect on reduced infarct size/AAR ratio. However, even the SNT- contains 1,5 µM S1P and has almost no effect on the infarct size/AAR ratio. Further experiments were done using SNT-/+ from S1P-deficient platelets. Indeed SPT- as well as SPT+ contained 70 % less S1P. Although SPT+ from S1P-deficient thrombocytes was not able to decrease the infarct/AAR ratio, the SNT- control from S1P-deficient thrombocytes (70 % lower S1P levels) showed an infarct size/AAR that is less than that of the S1P-containing SNT- control (Figure 1E).

The authors suggest that the S1P₁ receptor subtype is responsible for the protective effect of S1P. This protective effect is evident by a reduced infarct size and a higher left ventricular pressure gradient (DP). However, when they used W146 as an S1P₁ antagonist, DP was increased in the presence of W146, contradicting the proposed protective effect. This should be explained. However, it is well known that S1P acts via the S1P₃ on the heart rate, which should also be considered and affect DP.

The authors used a cardiomyocyte specific S1P₁ deficient mice to indicate that S1P₁ is responsible for the protective effect. However, the infarct size is comparable to the wildtype although even SNT- contains S1P. How is the S1P receptor profile in the myocardia when S1P₁ is abrogated?

The authors stated that the origin of increased S1P levels are due to an activation of platelets. To verify this, they have measured S1P levels in plasma, platelets, red blood cells and the myocardium. It is of interest how the S1P levels were measured in activated platelets. The material and method part describes the measurement of S1P in plasma. The measurement of S1P in platelets, cells and tissues has not been described in the manuscript.

Critically therapeutically, the authors also examined the effect of cangrelor and tirofiban on S1P release. They were able to show that cangrelor is able to inhibit S1P release from platelets. The effect of tirofiban was less pronounced. Cangrelor targets the platelet ADP receptor, whereas tirofiban is an inhibitor of the glycoprotein IIb/IIIa receptor. SNT+ contains less S1P when cangrelor is used. But it must be kept in mind that platelet activation was triggered by ADP alone in the experiments performed. Therefore, it would certainly make sense to use the two aggregation inhibitors if thrombocyte activation occurred in a different way.

The clinical data show that patients have an improved outcome when plasma S1P levels are

elevated. The study appears to be somewhat out of context, as it provides no indication of what the elevated S1P levels are attributed to. It would certainly be of interest to investigate whether S1P levels are different depending on whether one treats with cangrelor or tirofiban. Due to a dual therapy with ASS in the guidelines, this is certainly not possible.

Point-by-point response to the reviewers

We thank the reviewers for their constructive criticism and comments, and appreciate the opportunity to resubmit a revised version. We have addressed all points by a series of new experiments and provide all data in the revised manuscript and below on a point-by-point basis.

REVIEWER COMMENTS

Reviewer #1 (Remarks to the Author):

The manuscript by Polzin et al. sets out to define a novel cardioprotective role of platelet-derived sphingosine-1-phosphate (S1P) burst in acute myocardial infarction (AMI). The authors show that administration of supernatants from activated platelets reduced infarct size in a mouse model of AMI, which was not observed with platelets deficient for S1P export (Mfsd2B) or production (Sphk1) or in mice deficient for cardiomyocyte S1P receptor 1. Also in STEMI patients, S1P levels appeared to be inversely associated with cardiovascular mortality and infarct size.

The authors employ a straightforward approach to test the cardioprotective benefits of activated platelet supernatant on experimental MI in two different mouse models: (1) in vivo 30 min ischemia followed by 24h, 5 days or 21 days of reperfusion; (2) ex vivo retrograde cardiac perfusion (Langendorff model) and 40 min ischemia followed by 120 min reperfusion. Overall, the study proposes an interesting and innovative concept of translational relevance but currently several major concerns limit overall enthusiasm and would need to be addressed in order to corroborate the conclusions and to increase the impact of the findings.

Major comments:

1. The reduction of the infarcted area in Fig. 1A and Fig. 2C is moderate but significant. It is unclear whether the study and the number of mice used was subjected to a stringent power calculation, given the biological variation associated with the model. This should be critically evaluated to put the effectiveness of a single protective agent in context, but also to confidently assess the underlying mechanisms as postulated. Accordingly, this may help to unravel a significant effect on EF in Fig. 1A.

Answer: We completely agree and have increased the sample by n=4 (NaCl), n=5 (SNT-) and N=6(SNT+) now resulting in a total of n=13 per SNT-group and n=10 in the NaCl-group (Fig. 1A, 2C).

We have performed a stringent power analysis: Over the last 5 years we have gathered infarct size data from n=256 mice of identical background and age with the same protocol as in this manuscript. The infarct size there was $40.0 \pm 7.5\%$ (IS/AAR). The current infarct size in the NaCl group in the current manuscript was 41.3%, which is exactly in this range. Assuming that the bioactive compound in SNT+ is, indeed, S1P and based on our work and that by others that S1P protects against cardiac ischemia/reperfusion in vivo, we would expect a maximum reduction of infarct size of 28% (alpha error of 0.05 and a beta error of 0.2) according to the concentration kinetic by pure S1P we have established previously [1]. This results in an effect size r of 0.62 and a calculated n-number of n=10 to detect it. Accordingly, in our SNT+ experiments of the current paper we have observed a statistically significant reduction of infarct size of 24% with an actual n-number of n=13. Thank you for having us increase the statistical power.

2. In general, the group size $n=4-5$ for histology and echocardiography (Fig. 1A-C) is quite low, given the relatively small difference between the two experimental groups. As outlined above, this reviewer would hence recommend a higher n number to increase the data robustness. It is recommended to report functional echocardiography data over time, preferably including sham controls. Alternatively, the baseline and 24h echocardiography data could be provided in a supplemental table.

Answer: Thank you for your advice that we were happy to follow. As recommended, we have increased the number of experiments to increase data robustness according to the exact above calculations. We now also provide the complete functional echocardiography data over time including sham controls as recommended (now supplement Table 3).

3. With regards to the histology data in Fig. 1B, some overview images of cardiac cross-sections should be added to indicate the area of analysis for neutrophils and caspase 3. The time point at day 5 is beyond the neutrophil peak in the infarcted re-perfused heart, which occurs around 24 h post-ischemia.

Answer: We have now added the requested overview images of cardiac cross-sections to indicate the area of analysis. Also, we have omitted the time point at day 5 point and have analyzed the area for neutrophils at their peak at 24h as correctly indicated by the reviewer. Here, we see a clear reduction by ~50% in the SNT+ group compared to SNT- group (new Fig. 1B). We have also increased the number of experiments in our histological analysis of caspase 3-positive cells by $n=3$ now resulting in a total of $n=7$. Here, we have also observed a reduction by ~40% in the SNT+ group (new Fig. 1C).

4. Interestingly, there is a trend towards smaller MI when using Sphk1^{-/-} supernatants (Fig. 1D) even when not activated. Could this be interrogated for a reduction in other, possibly exacerbating factors or is this due to biological variation only? In general, the effects of Sphk1 deficiency are not convincing.

Answer: We understand and have looked into this in detail by increasing the sample size in the SphK1^{-/-} groups to make sure we do not oversee any other effects (new Fig. 1F). Accordingly, we can now exclude any smaller MI in the SNT- Sphk1^{-/-} supernatants compared to SNT- C57Bl6 control supernatants (new Fig. 1F). The increase in sample size now clearly supports the lack of difference between SNT- and SNT+ from Sphk1^{-/-} on infarct size.

5. Again, in Fig. 2C only small effects are observed and the number of mice per group with $n=4$ is not sufficient. Labelling is incomplete. Other cell types including RBCs should be shown in Fig. 2D.

Answer: We agree and have increased the number of mice per group to $n=6$. We apologize for the incomplete labelling; this has been now corrected. In addition, we now provide as requested data on S1P content of red blood cells and cardiac tissue in Fig. 2D. The lack of any changes there in contrast to major ones in platelets clearly supports that platelets are, indeed, the main source.

6. The exacerbating effect in mice lacking the platelet S1P exporter in Fig. 2E appears more pronounced than the protective effects with a lack of production in Fig. 1E. Could this be due to effects of the transporter on other (lipid) mediators?

Answer: This is an excellent observation that had eluded us until you the reviewer has drawn our attention. The question is rather tough to answer, though. One major difference between lack of exporter and lack of production is that lack of Mfsd2b results in 50-60-fold S1P accumulation in platelets, whereas lack of Sphk1 results in only ~2-fold less S1P in platelets. One could thus imagine that S1P export rather than production is rate limiting for the S1P “export burst” during MI. As for other mediators: this is possible but so far unknown. We have added these considerations to the discussion (page 9, lines 15-18). We appreciate the idea.

7. The manuscript would benefit from a better explanation why two different models were used, e.g. the Langendorff model may allow studying the cardioprotective effects without systemic immune cell responses. Indeed, the cardiomyocyte-specific S1PR KO model supports that the platelet-S1P effect on cardiomyocytes is involved in the reduction of infarct size. This might be further strengthened by co-staining of caspase 3 with cardiomyocytes in the infarct area. Currently, the resolution of the representative immunohistochemical images provided does not allow the reader to distinguish between positive and negative cells.

Answer: Thank you for this comment: yes, we have now done a better job in explaining the two models. This is, in fact, rather important for the understanding of the whole concept. Indeed, Langendorff experiments were performed to study local cardiac cardioprotection without interference by systemic immune cell responses. We have explained this in the manuscript (page 6, lines 16-17 and page 14, lines 2-3). Furthermore, we have increased the number of experiments in S1PR1 deficient animals as recommended by additional n=3 per group (new Figure 2C) and have also improved quality and resolution of the representative immunohistochemical images (neutrophils and caspase-3). Now cells and staining can be appreciated much better (new Figure 1B, C).

8. In Fig. 3A,B, the SNT- control is missing and there does not appear to be a significant difference between the cangrelor and tirofiban groups. Even though there is a trend, it can be questioned whether this would fully explain the effects in Fig. 3B, when taking into account the effects observed with a 70% reduction in S1P observed in Fig. 1D,E. How do the authors explain the high number of replicates in the vehicle group (n=15) as compared to the treatment groups in Fig. 3A, and to the experiments in Fig. 3B.

Answer: We agree that this needs to be worked out more precisely. We have thus included the requested SNT- group and have increased the numbers in all groups to test stringently for differences in infarct size. As shown below, we now find clear reduction of infarct size with SNT+ from the ADP and ADP+tirofiban groups and no rescue by SNT+ from ADP+cangrelor. We have also repeated the experiment in Fig. 3A to now show paired testing of all groups as the experiment was performed with all compounds in each individual (now Fig. 3 A,B).

Platelet S1P release during activation was reduced by P2Y12 inhibition (Can) but preserved during GPIIb/IIIa inhibition (Tiro, ANOVA-analysis, n=9). This translated to blunted cardioprotection by SNT+ of cangrelor treated platelets but sustained cardioprotection during tirofiban treatment (ANOVA analysis, n=8).

9. One limitation of the study is that only a single platelet agonist i.e. ADP was studied. The conclusions drawn, e.g. that inhibiting P2Y12 blocks the release of S1P from platelets may not be valid for other platelet activators contained in ruptured plaques such as collagen. This possibility should at least be acknowledged.

Answer: Thank you for this comment. This is a very important aspect that we decided to address experimentally. Thus we examined cardioprotection in mice that received SNT+ generated after platelet activation with collagen: this SNT was also cardioprotective (see figures below that were added to the manuscript and as supplemental figure 1; page 5 lines 25-26). We could also measure considerable amounts of S1P being released in the SNT+ from collagen-treated human platelets (supplemental figure 1).

10. Clinical data: the authors have evaluated a prospective cohort of 127 patients with acute STEMI with a 12-month follow-up for hard outcomes (cardiovascular death). However, some aspects are unclear and may influence the interpretation of the data, namely:

a. It is puzzling that the negative correlation in Fig. 3F appears to be largely driven by one patient. More importantly, can the authors exclude that the effect shown in Fig. 3E can be preserved in a multi-parametric analysis including lipid levels, as S1P may be a surrogate marker for HDL levels.

Answer: Thank you very much for this important comment. As suggested, we performed a multivariate analysis including lipid parameters and several procedural details (see also comment 10d). We found S1P levels to be the only robust predictor of left ventricular infarct size in our cohort (p=0.039). Lipid parameters and procedural details had no significant influence on infarct size.

Multivariable Regression Infarct Size

	Unstandardized beta	Standardized beta	p value
--	---------------------	-------------------	---------

Constant	49.133		0.000
Plasma S1P	-0.003	-0.380	0.039
LDL	0.144	0.546	0.401
HDL	0.103	0.143	0.608
Cholesterine	-0.253	-1.032	0.162
Triglycerides	0.019	0.130	0.640
Culprit Vessel	-1.228	-0.126	0.483
TIMI	3.738	0.198	0.309
Fluroscopy time	-0.219	-0.337	0.528
Amount contrast media	0.010	0.121	0.827
No. of stents	-0.536	-0.058	0.752
Symptom to balloon time	0.001	0.278	0.129

S1P = Sphingosin-1-Phosphate, LDL = low-density lipoprotein, HDL = high-density lipoprotein, TIMI = Thrombolysis in Myocardial Infarction, No = number

b. Blood sampling was performed on admission, and at 12 hours and 5 days after the intervention. However, data presented in Fig. 3E, F only refer to S1P concentration on admission. Do the authors observe time-dependent changes in plasma S1P (as in the mouse experiments)? Would S1P levels assessed after intervention still be predictive of mortality?

Answer: We thank the reviewer for this question. S1P concentrations did not differ significantly between time points. However, these were not as suitable to monitor changes over time as those from the precisely timed mouse experiments (this is surely due to the in “real world” different t=0 after the onset of clinical MI). To answer the second question: S1P levels at the other time points were not associated with mortality. We have added all this to the manuscript (page 8, lines 18-19 and page 10, lines 9-12).

c. Plasma S1P was evaluated at hospitalization for Fig. 3E, F and the authors state that “blood sampling was performed during ischemia” (lines 365-366). However, there is no mentioning of the interval between the occurrence of symptoms and the sampling (and the intervention). This is crucial, as the authors show the extremely fast and transient release of S1P from platelets in their mouse experiments (Fig. 2D) and as longer “symptom-to-balloon” may account for a worse prognosis in STEMI patients.

Answer: The reviewer is right that in general, symptom to balloon time is crucial for prognosis in patients undergoing primary PCI. Thus the multivariate analysis we now performed included symptom to balloon time (answer to comment 10a) and revealed that in our cohort, the symptom to balloon time did not have significant influence on infarct size ($p=0.129$). We added this to the manuscript (supplemental table 2).

d. The nature of the intervention (line 366) is unclear. This reviewer assumes that most (if not all) of the patients underwent primary PCI, but the number of affected/treated lesions, the use of DES or BMS, and the TIMI grade for revascularization are not specified. All these parameters may have a prognostic role in patients with AMI and should therefore be considered.

Answer: Indeed, procedural details are crucial in the outcome after PCI. We have now added several important procedural details (culprit vessel, contrast medium, TIMI flow, number of

stents) to our multivariate analysis (answer to comment 10a). None of these characteristics were predictive for the infarct size. All patients received drug-eluting stents, so the type of stent was not included in the analysis. We added this to our manuscript and we thank the reviewer for this question.

e. Fig. 3E only includes data for n=38 patients. The authors should report data for all patients in the cohort or explain why a considerable number of cases have been excluded from this analysis.

Answer: Thank you very much for this question. Indeed, a high number of patients was not eligible for cardiac magnetic resonance imaging due to different reasons such as implanted pacemaker (n=12), allergy against contrast medium (n=8) and several others. For a detailed presentation, we now provide the following flow chart in supplemental table 2.

f. The analyses in Fig. 3E, F are univariate, however, the authors may consider multivariate analyses to adjust the results for other prognostically relevant factors. Indeed, the lack of statistical significance for parameters in tables S1 and S2 may reflect an overall limited statistical power of the investigated cohort. In this regard, the methods paragraph does not report an a priori evaluation of the statistical power, which – ideally - should be provided.

Answer: We thank the reviewer for the suggestion to perform a multivariate analysis in this setting. We now performed analyzes before and after inverse probability treatment weighting as matching analysis (IPTW). High S1P concentrations were a robust predictor of death in our cohort (unweighted: p=0.04, IPTW: p=0.03). We added this to the results section and the following graphs and tables into the supplemental (page 8, lines 20-26).

Characteristics of included patients before IPTW

	High (N=42)	Medium (N=43)	Low (N=42)	p value
Characteristic				

Age	63.76 ± 13.70	62.28 ± 13.67	62.76 ± 13.53	.878
Male Gender	32 (76.2%)	31 (72.1%)	29 (69.0%)	.763
BMI	27.48 ± 4.57	28.36 ± 4.23	26.53 ± 5.23	.205
DM	9 (21.4%)	9 (20.9%)	10 (23.8%)	.943

Characteristics of included patients after IPTW

	High (N=42)	Medium (N=43)	Low (N=43)	p value
Characteristic				
Age	62.63 ± 13.74	63.22 ± 14.02	63.65 ± 12.77	.941
Male Gender	30 (71.4%)	30 (71.4%)	32 (74.4%)	.938
BMI	27.54 ± 4.70	27.73 ± 3.96	28.32 ± 6.35	.765
DM	9 (21.4%)	9 (21.4%)	10 (23.3%)	.973

g. It would be interesting to learn whether S1P may improve the reclassification of patients when tested over established risk algorithms (e.g. TIMI, GRACE risk score).

Answer: Yes, this is, indeed, interesting; thank you for pointing it out. To explore whether S1P concentration after myocardial infarction could improve the predictive power of the GRACE score, we first computed the GRACE score for each patient in our sample. We then allocated values to each of the three S1P concentrations. We started with the values 0, 1, and 2 for high, medium and low S1P concentration, respectively, and added them to the GRACE score of patients. We then computed an ROC curve to explore whether the AUC improved. We used Python programming to test each combination of values for high concentration in the range 0-20, medium concentration in the range 1-40 and low concentration in the range 2-60. Each combination was added to the patients GRACE score and an ROC curve was computed. The combination of values yielding the highest AUC in the ROC analysis was selected.

Adding the S1P concentration to the GRACE score with high concentration = 0 points, medium concentration = 19 points, and low concentration = 32 points, resulted in an AUC of 0.73. This was higher than the ROC AUC of 0.67 of the original GRACE score. Although this did not reach statistical significance to show that S1P improved the reclassification of patients we believe that this may still be the case if sample size would be increased. We will certainly focus on this in the future. Our current data show a p-value of 0.11 (see figure below).

A) ROC curve of original GRACE Score on 6-month mortality; B) ROC of new GRACE score including S1P concentration on 6-month mortality C) Pairwise comparison of A+B ROC curves

Minor comments:

1. Please add representative images of TTC staining in Fig. 1A. The analysis of apoptosis by staining for active caspase-3 in Fig. 1B does not reveal the cellular context and should be repeated by double immunofluorescence staining including cell-specific markers to distinguish resident from inflammatory cell types. In general, scale bars are missing.

Answer: We added representative images of TTC staining in Fig. 1A und new representative images of histological analyses in Fig. 1B, where cardiomyocytes are now morphologically distinguished. Scale bars are now included in Fig. 1B.

2. Was the observational study registered in any publicly available registry (e.g. ClinicalTrial.org)? If so, the registration ID should be provided.

Answer: The observational study is registered on clinicaltrials.org with the ID "NCT03539133". We added this to our manuscript (page 20, line 5).

3. Statistical analyses: (a) it is unclear whether homogeneity of variances has been tested, especially prior to ANOVA and how the authors dealt with heteroscedastic data; (b) it is unclear whether the authors have assessed normal distribution also before applying bivariate correlation tests; (c) "Tuckey" should rather read "Tukey".

Answer: We have revised the section "Statistical analyses" accordingly (page 18-19). In detail, we added that "homogeneity of variance was tested with Levene test (a). In case of heteroscedastic data, Welch test was performed. Comparison of three or more groups was performed using ANOVA-analyses. If the assumption of homogeneity of variances was not met, indicated by a significant Levene test, a Kruskal-Wallis test was performed. We have also specified that (b) "normality of distribution of variables used for Pearson correlation was assessed with Kolmogorov-Smirnov test. In case of not normally distributed variables Spearman correlation was performed". We have also corrected the spelling error (c) and replaced "Tuckey" with "Tukey".

4. In supplemental table 2, it is somewhat surprising that apparently none of the patients was pretreated with a P2Y12 inhibitor and therefore this has not listed. For completeness and because this plays an important role in this paper, this should nevertheless be stated.

Answer: We thank the reviewer for this comment. The reviewer is completely right that no patient was pre-treated with a P2Y12 inhibitor. This is due to the fact that loading with P2Y12 inhibitor happens at the time of the PCI in our center according to the "ESC guideline for the management of acute myocardial infarction in patients presenting with ST-segment elevation" from 2017 [2].

Reviewer #2 (Remarks to the Author):

The manuscript 'Revealing concealed cardioprotection by platelet Mfsd2b-released S1P in human and murine myocardial infarction' submitted by Polzin and colleagues indicate that activated platelets release sphingosine 1-phosphate during acute myocardial infarction, which is cardioprotective via stimulation the S1P1 receptor subtype.

The authors indicate that S1P is released from platelets after activation via the S1P exporter Mfsd2b and binds to the S1P1 receptor subtype, which is protective against ischemia injury. It is well known from several studies that administration of exogenous S1P protects against ischemia/reperfusion injury. Further studies indicate that S1P is released after thrombocyte activation via the Mfsd2b transporter. Thus, the novelty of the study is a platelet-mediated intrinsic cardioprotection.

There are some points in the manuscript that do not seem logical and would need to be addressed.

The authors used murine platelet supernatant with and without prior activation via ADP (SNT- and SNT+) to examine the effect on myocardial ischemia. They were able to show that SNT+ contained 25 % more S1P compared to SNT- and that SNT+ was able to decrease infarct size/AAR-ratio measured via computer-assisted planimetry. This experiment should demonstrate the S1P-mediated effect on reduced infarct size/AAR ratio. However, even the SNT- contains 1,5 μ M S1P and has almost no effect on the infarct size/AAR ratio. Further experiments were done using SNT-/+ from S1P-deficient platelets. Indeed, SNT- as well as SNT+ contained 70 % less S1P. Although SNT+ from S1P deficient thrombocytes was not able to decrease the infarct/AAR ratio, the SNT- control from S1P-deficient thrombocytes (70 % lower S1P levels) showed an infarct size/AAR that is less than that of the S1P-containing SNT-control (Figure 1E).

Answer: This is, indeed, a highly relevant question with vast implications. To answer it in depth, we have generated a dose-response relationship between infarct size reduction and S1P concentration that is based on our own data sets over the years (n=7 with a total of n=96 mice of the same background, age and sex and using the identical I/R model [1, 3]). The data are presented in the figure below. There, we see a sigmoidal dose-response between S1P and reduction of infarct size with a rather narrow slope range. Within this slope, small increases in plasma S1P have a pronounced effect on cardioprotection. Lower S1P levels show no cardioprotection, whereas larger increases in S1P have no further benefit. This may also explain the lack of difference between SNT- from S1P-deficient thrombocytes and C57Bl6 controls despite the difference in S1P concentrations as correctly noted by the reviewer (new Figure 1E). We have added this in supplemental Figure 3 and in the discussion on page 9, lines 15-16. Thank you for having us do this.

A sigmoidal dose-response curve between plasma S1P and reduction of infarct size after I/R shows a narrow slope range. The figure is based on $n=7$ individual data sets with a total of $n=96$ mice in the same model [1, 3]. Studies in detail from left to right: (1) untreated mice, SNT- and SNT+ treated mice from this study (red and green dots); (2, 3, 4) mice treated with 3,8 ng/g, 19 ng/g, and 38ng/g S1P administered intravenously 15 min prior to AMI [1]; (5) mice treated with S1P lyase inhibitor DOP seven days prior AMI resulting in an endogenous increase of plasma S1P as measured by LC/MS [3].

The authors suggest that the S1P1 receptor subtype is responsible for the protective effect of S1P. This protective effect is evident by a reduced infarct size and a higher left ventricular pressure gradient (DP). However, when they used W146 as an S1PR1 antagonist, DP was increased in the presence of W146, contradicting the proposed protective effect. This should be explained. However, it is well known that S1P acts via the S1PR3 on the heart rate, which should also be considered and affect DP.

Answer: The reviewer is correct: We had overseen this apparent discrepancy, which turned out to be due to sample size. After increasing the sample size in each group (additional two mice in each group), we can now exclude any differences between. The data is presented below and in a revised figure 2A. As to the question on heart rate: In the Langendorff experiments, the heart rate is experimentally set at exactly 600bpm. Thus biases due to heart rate are excluded.

SNT+ leads to decreased infarct size (IS [%LV]), higher left ventricular pressure gradient (DP) and lower left ventricular end-diastolic pressure (LVEDP) in a model of Langendorff-perfused hearts. This was blunted by pharmacological inhibition of S1P receptor 1 (S1PR1; Two-way ANOVA, Vehicle $n_{SNT-}=10$, $n_{SNT+}=8$, W146 $n_{SNT-}=7$, $n_{SNT+}=5$).

The authors used a cardiomyocyte specific S1PR1 deficient mice to indicate that S1PR1 is responsible for the protective effect. However, the infarct size is comparable to the wildtype although even SNT- contains S1P. How is the S1P receptor profile in the myocardia when S1PR1 is abrogated?

Answer: We see no compensatory changes in the expression of S1PR2 and S1PR3 in S1PR1 cardio cre+ compared to cre- (data below and now on page 6, lines 25-26). The data is also provided in our original paper on the characterization of the S1PR1 cardio cre+ mouse [4]. Meanwhile, as a reviewer from another study also requested original data, we have now published them also in our most recent paper [5].

The authors stated that the origin of increased S1P levels is due to an activation of platelets. To verify this, they have measured S1P levels in plasma, platelets, red blood cells and the myocardium. It is of interest how the S1P levels were measured in activated platelets. The material and method part describes the measurement of S1P in plasma. The measurement of S1P in platelets, cells and tissues has not been described in the manuscript.

Answer: We have now done this in detail in the manuscript (page 12-13). Here the excerpt:

“Platelets were resuspended in MeOH, followed by addition of 10 μ L internal standards (for S1P (d18:1): 10 pmol C₁₇ S1P in MeOH; for S1P(d17:1): d₇ S1P in MeOH, Avanti Polar Lipids Inc., Alabaster, AL). Samples were mixed, precipitated overnight at -80°C and centrifugated (5 min at 21300 rcf, 4°C). The supernatant was transferred into mass spectrometry sample vials and stored at -80°C until measurement. Tissue samples (10-20 mg were homogenized in a Stomacher Bag with 10 μ L internal standard solution and 500 μ L MeOH. The homogenate was collected, another aliquot MeOH (500 μ L) was added to the stomacher bag and, after rinsing, the solution was pooled with the first homogenate (total 1 mL). Samples were mixed, precipitated overnight at -80°C and centrifugated (5 min at 21300 rcf, 4°C). The supernatant was transferred into mass spectrometry sample vials and stored at -80°C until measurement. Chromatographic separation was performed on a LCMS-8050 triple-quadrupole mass spectrometer (Shimadzu Duisburg, Germany) interfaced with a Dual Ion Source and a Nexera X3 Front-End-System (Shimadzu Duisburg, Germany). HPLC is performed with a 2x60 mm MultoHigh-C18 RP column with 3 μ m particle size at 40°C . Mobile phases consisted of [A] 1% aq. HCO₂H and [B] MeOH. The following LC gradient was used: Start at 10% [B], linear increase of [B] from 10% to 100% from 0 min to 3.0 min (B-Curve = -2), 100% [B] from 3.00 min to 8 min and equilibration from 8.01 min to 10.00 min prior next injection. Flow rate was 400 μ L/min and injection volume of all samples was 10 μ L. MS settings were the following: Interface: ESI, Nebulizing Gas flow: 2 L/min, Heating Gas Flow: 10 L/min, Interface Temperature: 300°C , DL Temperature: 250°C , Heat Block Temperature: 400°C , Drying Gas Flow: 10 L/min. Data were collected using multiple reaction monitoring (MRM). Positive ionization was used for qualitative analysis and quantification. Standard curves were generated by measuring increased amounts (100 fmol to 50 pmol) of S1P (d18:1) and C₁₇

S1P (d17:1) together with internal standard (C_{17} S1P (d17:1) for S1P (d18:1); d_7 S1P (d18:1) for C_{17} S1P (d17:1), 0.3 μM final conc. in MeOH). Linearity of the standard curves and correlation coefficients were obtained by linear regression analysis. All MS analyses were performed using LabSolutions 5.99 SP2 and LabSolutions Insight (Shimadzu Duisburg, Germany).”

Critically therapeutically, the authors also examined the effect of cangrelor and tirofiban on S1P release. They were able to show that cangrelor is able to inhibit S1P release from platelets. The effect of tirofiban was less pronounced. Cangrelor targets the platelet ADP receptor, whereas tirofiban is an inhibitor of the glycoprotein IIb/IIIa receptor. SNT+ contains less S1P when cangrelor is used. But it must be kept in mind that platelet activation was triggered by ADP alone in the experiments performed. Therefore, it would certainly make sense to use the two aggregation inhibitors if thrombocyte activation occurred in a different way.

Answer: The reviewer is absolutely right, and we have now performed platelet activation with collagen. This is a very important aspect that we have decided to address experimentally. Thus, we examined cardioprotection in mice that received SNT+ generated after platelet activation with collagen and found it to be cardioprotective as well (see figure below, now supplemental figure 1A and provided in the manuscript on page 5, lines 25-26). Furthermore, we also measured considerable amounts of S1P being released in the SNT+ from collagen-treated human platelets (figure below, now supplemental figure 1A; rightest panel). Finally, we have observed that tirofiban but not cangrelor preserved S1P release (see figure below; data added as supplemental figure 1B and provided on page 8, line 5).

Injection of cell-free supernatant of collagen (10 $\mu\text{g}/\text{ml}$)-activated platelets (SNT+) prior to AMI lead to decreased infarct size after 24 h of reperfusion as compared to supernatant of non-activated platelets (SNT-) (t-test, n=7). Echocardiographic assessment 24 h post AMI showed improved cardiac function in SNT+ treated mice (t-test, n=7). S1P content in SNT from collagen-treated human platelets is increased compared to SNT of unstimulated platelets as measured by LC/MS (paired t-test, n=11).

Platelet S1P release during activation by collagen (10 $\mu\text{g}/\text{ml}$) was reduced by P2Y₁₂ inhibition (Can) but preserved by GPIIb/IIIa inhibition (Tiro, ANOVA-analysis, n=5). Blood was taken from donor mice via retrobulbar plexus with a heparinized cannula and rested for 15 minutes at 37°C. After first centrifugation (170 g, 5 min, room temperature, RT)) platelet rich plasma (PRP) was taken and incubated at 37°C with 10 $\mu\text{g}/\text{ml}$ collagen (Probe & go Labor Labordiagnostica GmbH, Germany). After 3 min, SNT+ was generated via second centrifugation (2000 g, 5 min, RT). Inhibitors were added after first centrifugation and 2 min prior to ADP incubation (200nM cangrelor, 50nM tirofiban).

The clinical data show that patients have an improved outcome when plasma S1P levels are elevated. The study appears to be somewhat out of context, as it provides no indication of what the elevated S1P levels are attributed to. It would certainly be of interest to investigate whether S1P levels are different depending on whether one treats with cangrelor or tirofiban. Due to a dual therapy with ASS in the guidelines, this is certainly not possible.

Answer: We thank the reviewer for this comment. Our experimental data indicate that platelets are responsible for the elevated S1P during AMI. However, the reviewer is absolutely correct: we cannot prove that platelets are also the S1P source in our patients unless we compare patients treated with P2Y12 or gpIIb/IIIa inhibitors. However, due to recent guidelines [2] this approach cannot be undertaken. We believe that our data (if, indeed, the S1P source in humans is identical as in mice) may theoretically support a treatment regime where early gpIIb/IIIa inhibition followed by a later initiation of P2Y12 inhibition may improve myocardial remodeling after AMI. This has to be evaluated in further trials and is certainly an exciting and potentially relevant clinical perspective.

References:

1. Theilmeyer, G., et al., *High-density lipoproteins and their constituent, sphingosine-1-phosphate, directly protect the heart against ischemia/reperfusion injury in vivo via the S1P3 lysophospholipid receptor*. *Circulation*, 2006. **114**(13): p. 1403-9.
2. Ibanez, B., et al., *2017 ESC Guidelines for the management of acute myocardial infarction in patients presenting with ST-segment elevation: The Task Force for the management of acute myocardial infarction in patients presenting with ST-segment elevation of the European Society of Cardiology (ESC)*. *Eur Heart J*, 2018. **39**(2): p. 119-177.
3. Polzin, A., et al., *S1P Lyase Inhibition Starting After Ischemia/Reperfusion Improves Postischemic Cardiac Remodeling*. *JACC Basic Transl Sci*, 2022. **7**(5): p. 498-499.
4. Keul, P., et al., *Sphingosine-1-Phosphate Receptor 1 Regulates Cardiac Function by Modulating Ca²⁺ Sensitivity and Na⁺/H⁺ Exchange and Mediates Protection by Ischemic Preconditioning*. *J Am Heart Assoc*, 2016. **5**(5).
5. Polzin, A., et al., *Sphingosine-1-phosphate improves outcome of no-reflow acute myocardial infarction via sphingosine-1-phosphate receptor 1*. *ESC Heart Fail*, 2022.

REVIEWER COMMENTS

Reviewer #1 (Remarks to the Author):

The authors are to be commended for an excellent and thorough revision addressing all my concerns.

Reviewer #2 (Remarks to the Author):

Answers to the criticism point 1, I don't completely agree with the authors. They suggest a sigmoidal dose-response curve between plasma S1P and reduction of infarct size after I/R. The data are based from several experiments. When S1P is administered in a dose response (2-4), which is very similar to the SNT treatment, it is visible that there is a dose-dependent reduction of the IS. The sigmoid curve is only a result of including data from the S1P lyase inhibitor. But it should be considered that this is an artificial treatment leading to an complete immune dysregulation. Thus, these data should not be included.

There are further ambiguities that do not seem logical after doing the additional experiments.

For most experiments with SNT(-) and SNT(+) they used 7, 8 or 13 preparations. In Fig 1 it is shown that Caspase activity and neutrophil infiltration are diminished in all SNT(+) preparations compared to the nonactivated supernatant. However, several SNT(-) samples show higher S1P amounts than activated supernatant. Thus, it seems likely that other factors released from SNT contribute to these effects.

Additionally, in Fig 1E the S1P content is shown. It is shown that the S1P content is increased approximately of about 25 %. Raw data in the Excel Table indicate that for the SNT(-) 16 samples were included and only 8 samples of the SNT(+). The S1P content was not measured for the SNT(+) preparations. If you compare the 8 SNT(-) with the corresponding SNT(+) values only an increase of 10 % is visible.

The authors stated that administration of supernatants from activated platelets reduced infarct size in murine AMI, which was blunted in platelets deficient for S1P export (Mfsd2B) or production (Sphk1) and in mice deficient for cardiomyocyte S1P receptor 1. The Mfsd2B-SNT experiment is missing indicating that the supernatant of Mfsd2B deficient thrombocytes is not able to have a cardioprotective effect. Instead, AMI was done in Mfsd2B deficient mice. However, it is well known that these mice possess a very low S1P plasma level. It must be considered that the MFsd2B is also present in erythrocytes, which are mainly responsible for the S1P content.

The authors indicate five minutes after ischemia an increased S1P level in the peripheral circulation in plasma, mirrored by an 80% decrease of platelet S1P in the same blood sample (Figure 2D). In contrast, RBC did not show any changes suggesting that the increase in circulating S1P during AMI may stem from platelet activation. The level of S1P in platelets is presented as S1P/ml blood (nmol), the S1P level in RBC in μM , which is not a unit for cells. It seems likely that the RBC S1P level is also S1P/ml blood (nmol). This is a 30.000-50.000 higher S1P content per ml blood compared to platelets. This means that a possibly non-significant release from erythrocytes could have an extremely large effect. And it must be noted that erythrocytes are also involved in thrombus formation. The raw data in Excel table 2D show only 3 measurements in platelets, which is not consistent with the legend to Fig. 2.

ANSWER TO REVIEWER COMMENTS

Reviewer #1 (Remarks to the Author):

The authors are to be commended for an excellent and thorough revision addressing all my concerns.

We very much appreciate this.

Reviewer #2 (Remarks to the Author):

We thank the reviewer for the thorough reassessment. We agree with all raised points and are happy to provide a new series of experiments to address the remaining points. These include experiments with Mfsd2b mice, new calculations and in depth discussion of all raised points.

Answers to the criticism point 1, I don't completely agree with the authors. They suggest a sigmoidal dose-response curve between plasma S1P and reduction of infarct size after I/R. The data are based from several experiments. When S1P is administered in a dose response (2-4), which is very similar to the SNT treatment, it is visible that there is a dose-dependent reduction of the IS. The sigmoid curve is only a result of including data from the S1P lyase inhibitor. But it should be considered that this is an artificial treatment leading to a complete immune dysregulation. Thus, these data should not be included.

We agree and have looked carefully into the data and removed as suggested the S1P lyase inhibitor data (this approach is, indeed, unlike directly applying S1P). We found the dose response to still remain sigmoidal as the doubling of the S1P dose from 19 to 38 ng/g increased the effect not by 100 but 40%. Actually, the fit is now even better. Please compare revised supplemental figure 3. Apart from that, we completely agree that the S1P lyase inhibitor may have other effects e.g. on immune cells but apparently, there is a major effect on cardiomyocytes as the benefit of DOP treatment was abolished in cardiomyocyte-specific S1PR1 Cardio Cre+ mice as we have recently published.

Polzin A, Dannenberg L, Benkhoff M, Barcik M, Keul P, Helten C, Zeus T, Kelm M, Levkau B. S1P Lyase Inhibition Starting After Ischemia/Reperfusion Improves Postischemic Cardiac Remodeling. *JACC Basic Transl Sci.* 2022 May 23;7(5):498-499. doi: 10.1016/j.jacbts.2022.03.009. PMID: 35663625; PMCID: PMC9156442.

There are further ambiguities that do not seem logical after doing the additional experiments.

For most experiments with SNT(-) and SNT(+) they used 7, 8 or 13 preparations. In Fig 1 it is shown that Caspase activity and neutrophil infiltration are diminished in all SNT(+) preparations compared to the nonactivated supernatant. However, several SNT(-) samples show higher S1P amounts than activated supernatant. Thus, it seems likely that other factors released from SNT contribute to these effects.

We understand the reviewer's point very well: in essence, the reviewer wonders why there is an overlap in S1P values in SNT- and SNT+ but no overlap in caspase activity or neutrophils. However, in the most crucial readout – infarct size (Fig 1 A) – there is an overlap as the reviewer would have expected and a highly significant infarct-reducing effect ($p=0.0024$). The most plausible explanation is that not all experimental readouts of the same phenomenon always match entirely.

Additionally, in Fig 1E the S1P content is shown. It is shown that the S1P content is increased approximately of about 25 %. Raw data in the Excel Table indicate that for the SNT(-) 16 samples were included and only 8 samples of the SNT(+). The S1P content was not measured for the SNT(+) preparations. If you compare the 8 SNT(-) with the corresponding SNT(+) values only an increase of 10 % is visible.

The reviewer's calculations are correct assuming that the values were matched. However, they were not, and this is why we had not used paired testing. We apologize for not stating this explicitly.

The authors stated that administration of supernatants from activated platelets reduced infarct size in murine AMI, which was blunted in platelets deficient for S1P export (Mfsd2B) or production (Sphk1) and in mice deficient for cardiomyocyte S1P receptor 1. The Mfsd2B-SNT experiment is missing indicating that the supernatant of Mfsd2B deficient thrombocytes is not able to have a cardioprotective effect. Instead, AMI was done in Mfsd2B deficient mice. However, it is well known that these mice possess a very low S1P plasma level. It must be considered that the MFsd2B is also present in erythrocytes, which are mainly responsible for the S1P content.

We agree and yes, this is a very comprehensive experiment that was, indeed, missing. We have performed experiments to address it and were happy to see them to line up with our work. The results are provided on page 7 and in figures 1 and 2:

“The S1P transporter Mfsd2b has recently been shown to be responsible for S1P release from activated platelets [38, 39]. We thus generated SNT+ and SNT- from Mfsd2b deficient mice and observed that SNT+ was unable to decrease infarct size (Figure 2E). Finally, we asked whether endogenous release of platelet-derived S1P during AMI by Mfsd2b might be cardioprotective: a scenario that most closely mimics the situation in patients. Remarkably, Mfsd2b^{-/-} mice had a 39% larger infarct size compared to Mfsd2b^{+/+} after AMI in vivo (Figure 2F). We excluded cardiac cell-specific effects of Mfsd2b, as infarct sizes were similar between in isolated Mfsd2b^{-/-} and Mfsd2b^{+/+} hearts in the Langendorff model (Figure 2G).”

The authors indicate five minutes after ischemia an increased S1P level in the peripheral circulation in plasma, mirrored by an 80% decrease of platelet S1P in the same blood sample (Figure 2D). In contrast, RBC did not show any changes suggesting that the increase in circulating S1P during AMI may stem from platelet activation. The level of S1P in platelets is presented as S1P/ml blood (nmol), the S1P level in RBC in μM , which is not a unit for cells. It seems likely that the RBC S1P level is also S1P/ml blood (nmol). This is a 30.000-50.000 higher S1P content per ml blood compared to platelets. This means that a possibly non-significant release from erythrocytes could have an extremely large effect. And it must be noted that erythrocytes are also involved in thrombus formation. The raw data in Excel table 2D show only 3 measurements in platelets, which is not consistent with the legend to Fig. 2.

We completely agree and have now recalculated the S1P units to enable comparison. The data on S1P in erythrocytes in Fig. 2D represents μmol S1P per liter packed but intact RBC. We now present the same data as “S1P per 10^{10} erythrocytes/ml (μM)” which is identical in units and numerically in the same range as in ref 38.

38. Vu, T.M., et al., *Mfsd2b is essential for the sphingosine-1-phosphate export in erythrocytes and platelets*. Nature, 2017. **550**(7677): p. 524-528.

The reviewer is correct that there is more S1P in the red blood cell compartment compared to the platelet compartment per ml blood. However, the difference is not 30.000-50.000 but 53-64 fold when calculated from above numbers. The calculation goes as follows: S1P contained in platelets per ml blood is ~25-30 pmol, whereas the 0.16-0.17 pmol S1P/10⁶ erythrocytes that we measure corresponds to 1600 nmol S1P in erythrocytes per ml blood.

And, yes, non-significant differences in release from erythrocytes may still have an effect on overall S1P levels. However, our LC-MS/MS lipidomics platform quantitatively covers an absolute range of 0.1 to 30 pmol/L S1P. Thus we are confident that we would detect such differences, and that the lack of changes in RBC S1P levels are not due to measurement artefacts or lack of sensitivity.

We have corrected the figure legend as indicated from n=4 to n=3 as correctly noticed by the reviewer.

All points raised were absolutely important and in need of clarification, and we thank the reviewer for giving us the chance to do this.

REVIEWERS' COMMENTS

Reviewer #2 (Remarks to the Author):

The authors have performed additional experiments to conclude that cardioprotection is due to a release of S1P from thrombocytes. Especially, the experiment generating SNT+ and SNT- from Mfsd2b deficient mice contributes to the quality of the manuscript. Moreover, the correct calculation of S1P levels in erythrocytes is also very important for the presented data. Thus, finally all points raised were clarified in the revised version of the manuscript.

Point-by-point response to the reviewers

REVIEWER COMMENTS

Reviewer #2 (Remarks to the Author):

The authors have performed additional experiments to conclude that cardioprotection is due to a release of S1P from thrombocytes. Especially, the experiment generating SNT+ and SNT- from Mfsd2b deficient mice contributes to the quality of the manuscript. Moreover, the correct calculation of S1P levels in erythrocytes is also very important for the presented data. Thus, finally all points raised were clarified in the revised version of the manuscript.

We are happy that our experiment generating SNT+ and SNT- from Mfsd2b deficient mice and the calculation of S1P levels in erythrocytes could clarify all open points. We very much appreciate this.